# The Skellam Mechanism for Differentially Private Federated Learning

**Naman Agarwal**[†]     **Peter Kairouz**[†]     **Ziyu Liu**[‡*]
[†]Google Research     [‡]Carnegie Mellon University
{namanagarwal, kairouz}@google.com, ziyuliu@cs.cmu.edu

## Abstract

We introduce the multi-dimensional Skellam mechanism, a discrete differential privacy mechanism based on the difference of two independent Poisson random variables. To quantify its privacy guarantees, we analyze the privacy loss distribution via a numerical evaluation and provide a sharp bound on the Rényi divergence between two shifted Skellam distributions. While useful in both centralized and distributed privacy applications, we investigate how it can be applied in the context of federated learning with secure aggregation under communication constraints. Our theoretical findings and extensive experimental evaluations demonstrate that the Skellam mechanism provides the same privacy-accuracy trade-offs as the continuous Gaussian mechanism, even when the precision is low. More importantly, Skellam is closed under summation and sampling from it only requires sampling from a Poisson distribution – an efficient routine that ships with all machine learning and data analysis software packages. These features, along with its discrete nature and competitive privacy-accuracy trade-offs, make it an attractive practical alternative to the newly introduced discrete Gaussian mechanism.

## 1   Introduction

The Gaussian mechanism is the workhorse for a multitude of differentially private learning algorithms [46, 10, 1]. While simple enough for mathematical reasoning and privacy accounting analyses, its continuous nature presents a number of challenges in practice. For example, it cannot be exactly represented on finite computers, making it prone to numerical errors that can break its privacy guarantees [35]. Moreover, it cannot be used in distributed learning settings with cryptographic multi-party computation primitives involving modular arithmetic, such as secure aggregation [12, 11]. To address these shortcomings, the binomial and (distributed) discrete Gaussian mechanisms were recently introduced [19, 2, 15, 25]. Unfortunately, both have their own drawbacks: the privacy loss for the binomial mechanism can be infinite with a non-zero probability, and the discrete Gaussian: (a) is not closed under summation (i.e. sum of discrete Gaussians is not a discrete Gaussian), complicating analysis in distributed settings and leading to a performance worse than continuous Gaussian in the highly distributed, low-noise regime [25]; (b) requires a sampling algorithm that is not shipped with mainstream machine learning or data analysis software packages, making it difficult for engineers to use it in production settings (naïve implementations may lead to catastrophic privacy errors).

**Our contributions**   To overcome these limitations, we introduce and analyze the multi-dimensional Skellam mechanism, a mechanism based on adding noise distributed according to the difference of two independent Poisson random variables. The Skellam noise is closed under summation (i.e. sums of Skellam random variables is again Skellam distributed) and can be sampled from easily – efficient Poisson samplers are widely available in numerical software packages. Being discrete in nature also

---

[*]Work done while at Google Research. [§]Alphabetical authorship.

35th Conference on Neural Information Processing Systems (NeurIPS 2021).

means that it can mesh well cryptographic protocols and can lead to communication savings.

To analyze the privacy guarantees of the Skellam mechanism and compare it with other mechanisms, we provide a numerical evaluation of the privacy loss random variable and prove a sharp bound on the Rényi divergence between two shifted Skellam distributions. Our careful analysis shows that for a multi-dimensional query function with $\ell_1$ sensitivity $\Delta_1$ and $\ell_2$ sensitivity $\Delta_2$, the Skellam mechanism with variance $\mu$ achieves $(\alpha, \varepsilon(\alpha))$ Rényi differential privacy (RDP) [36] for $\varepsilon(\alpha) \leq \frac{\alpha \Delta_2}{2\mu} + \min\left(\frac{(2\alpha-1)\Delta_2 + 6\Delta_1}{4\mu^2}, \frac{3\Delta_1}{2\mu}\right)$ (see Theorem 3.5). This implies that the RDP guarantees are at most $1 + O\left(1/\mu\right)$ times worse than those of the Gaussian mechanism.

To analyze the performance of the Skellam mechanism in practice, we consider a differentially private and communication constrained federated learning (FL) setting [26] where the noise is added locally to the $d$-dimensional discretized client updates that are then summed securely via a cryptographic protocol, such as secure aggregation (SecAgg) [11, 12]. We provide an end-to-end algorithm that appropriately discretizes the data and applies the Skellam mechanism along with modular arithmetic to bound the range of the data and communication costs before applying SecAgg.

We show on distributed mean estimation and two benchmark FL datasets, Federated EMNIST [14] and Stack Overflow [8], that our method can match the performance of the continuous Gaussian baseline under tight privacy and communication budgets, despite using generic RDP amplification via sampling [51] for our approach and the precise RDP analysis for the subsampled Gaussian mechanism [37]. Our method is implemented in TensorFlow Privacy [32] and TensorFlow Federated [24] and will be open-sourced.[2] While we mostly focus on FL applications, the Skellam mechanism can also be applied in other contexts of learning and analytics, including centralized settings.

**Related work**  The Skellam mechanism was first introduced in the context of computational differential privacy from lattice-based cryptography [49] and private Bayesian inference [45]. However, the privacy analyses in the prior work do not readily extend to the multi-dimensional case, and they give direct bounds for pure or approximate DP which makes only advanced composition theorems [28, 22] directly applicable in learning settings where the mechanism is applied many times. For example, the guarantees from [49] lead to poor accuracy-privacy trade-offs as demonstrated in Fig. 1. Moreover, we show in Section 3.1 that extending the direct privacy analysis to the multi-dimensional setting is non-trivial because the worst-case neighboring dataset pair is unknown in this case. For these reasons, our tight privacy analysis via a sharp RDP bound makes the Skellam mechanism practical for learning applications for the first time. These guarantees (almost) match those of the Gaussian mechanism and allow us to use generic RDP amplification via subsampling methods [51].

The closest mechanisms to Skellam are the binomial [2, 19] and the discrete Gaussian mechanisms [15, 25]. The binomial mechanism can (asymptotically) match the continuous Gaussian mechanism (when properly scaled). However, it does not achieve Rényi or zero-concentrated DP [36, 13] and has a privacy loss that can be infinite with a non-zero probability, leading to catastrophic privacy failures. The discrete Gaussian mechanism yields Rényi DP and can be applied to distributed settings [25], but it requires a sampling algorithm that is not yet available in data analysis software packages despite being explored in the lattice-based cryptography community (e.g., [43, 18, 38]). The discrete Gaussian is also not closed under summation and the divergence can be large in highly distributed low-noise settings (e.g. quantile estimation [6] and federated analytics [42]), which causes privacy degradation. See the end of Section 4 for more discussion.

## 2   Preliminaries

We begin by providing a formal definition for $(\varepsilon, \delta)$-differential privacy (DP) [20].

**Definition 2.1** (Differential Privacy). *For $\varepsilon, \delta \geq 0$, a randomized mechanism $M$ satisfies $(\varepsilon, \delta)$-DP if for all neighboring datasets $D, D'$ and all $\mathcal{S}$ in the range of $M$, we have that*

$$P\left(M(D) \in \mathcal{S}\right) \leq e^\varepsilon P\left(M(D') \in \mathcal{S}\right) + \delta,$$

*where $D$ and $D'$ are neighboring pairs if they can be obtained from each other by adding or removing all the records that belong to a particular user.*

In our experiments we consider *user-level differential privacy* – i.e., $D$ and $D'$ are neighboring pairs

---

[2]https://github.com/google-research/federated/tree/master/distributed_dp

if one of them can be obtained from the other by adding or removing *all* the records associated with a single user [33]. This is stronger than the commonly-used notion of item level privacy where, if a user contributes multiple records, only the addition or removal of one record is protected.

We also make use of Rényi differential privacy (RDP) [36] which allows for tight privacy accounting.

**Definition 2.2** (Rényi Differential Privacy). *A mechanism $M$ satisfies $(\alpha, \varepsilon)$-RDP if for any two neighboring datasets $D, D'$, we have that $D_\alpha(M(D), M(D')) \leq \varepsilon$ where $D_\alpha(P, Q)$ is the Rényi divergence between $P$ and $Q$ and is given by*

$$D_\alpha(P, Q) \triangleq \frac{1}{\alpha - 1} \log \left( \mathbb{E}_{x \sim Q} \left[ \left( \frac{P(x)}{Q(x)} \right)^\alpha \right] \right) = \frac{1}{\alpha - 1} \log \left( \mathbb{E}_{x \sim P} \left[ \left( \frac{P(x)}{Q(x)} \right)^{\alpha - 1} \right] \right).$$

A closely related privacy notion is zero-concentrated DP (zCDP) [21, 13]. In fact, $\frac{1}{2}\varepsilon^2$-zCDP is equivalent to simultaneously satisfying an infinite family of RDP guarantees, namely $(\alpha, \frac{1}{2}\varepsilon^2\alpha)$-Rényi differential privacy for all $\alpha \in (1, \infty)$. The following conversion lemma from [13, 15, 7] relates RDP to $(\varepsilon, \delta)$-DP.

**Lemma 2.3.** *If $M$ satisfies $(\alpha, \varepsilon)$-RDP, then, for any $\delta > 0$, $M$ satisfies $(\varepsilon_{DP}(\delta), \delta)$-DP, where*

$$\varepsilon_{DP}(\delta) = \inf_{\alpha > 1} \varepsilon + \frac{\log(1/\alpha\delta)}{\alpha - 1} + \log(1 - 1/\alpha).$$

For any query function $f$, we define the $\Delta_p$ sensitivity as $\max_{D, D'} \|f(D) - f(D')\|_p$, where $D$ and $D'$ are neighboring pairs differing by adding or removing all the records from a particular user. We also include the RDP guarantees of the discrete Gaussian mechanism (same RDP guarantees as the continuous Gaussian mechanism) to which we compare our method.

**Definition 2.4** (The Discrete Gaussian Mechanism [15]). *Given an integer-valued query $f(D) \in \mathbb{Z}^d$ and noise variance $\mu$, the Discrete Gaussian (DGaussian) Mechanism is given by*

$$f(D) + Z, \text{ where } Z \sim \mathcal{N}_{\mathbb{Z}}(0, \mu),$$

*and $\mathcal{N}_{\mathbb{Z}}(0, \mu)$ denotes the discrete Gaussian distribution defined in Equation (1) of [15]. The discrete Gaussian mechanism achieves $(\alpha, \frac{\alpha\Delta_2^2}{2\mu})$-Rényi DP.*

## 3 The Skellam Mechanism

We begin by presenting the definition of the Skellam distribution, which is the basis of the Skellam Mechanism for releasing integer ranged multi-dimensional queries.

**Definition 3.1** (Skellam Distribution). *The multidimensional Skellam distribution $\mathrm{Sk}_{\Delta, \mu}$ over $\mathbb{Z}^d$ with mean $\Delta \in \mathbb{Z}^d$ and variance $\mu$ is given with each coordinate $X_i$ distributed independently as*

$$X_i \sim \mathrm{Sk}_{\Delta_i, \mu} \text{ with } P(X_i = k) = e^{-\mu} I_{k - \Delta_i}(\mu),$$

for $k \in \mathbb{Z}$. Here, $I_\nu(x)$ is the modified Bessel function of the first kind. A key property of Skellam random variables which motivates their use in DP is that they are closed under summation, i.e. let $X_1 \sim \mathrm{Sk}_{\Delta_1, \mu_1}$ and $X_2 \sim \mathrm{Sk}_{\Delta_2, \mu_2}$ then $X_1 + X_2 \sim \mathrm{Sk}_{\Delta_1 + \Delta_2, \mu_1 + \mu_2}$. This follows from the fact that a Skellam random variable $X$ can be obtained by taking the difference between two independent Poisson random variables with means $\mu$.[3] We are now ready to introduce the Skellam Mechanism.

**Definition 3.2** (The Skellam Mechanism). *Given an integer-valued query $f(D) \in \mathbb{Z}^d$, we define the Skellam Mechanism as*

$$\mathrm{Sk}_{0, \mu}(f(D)) = f(D) + Z, \text{ where } Z \sim \mathrm{Sk}_{0, \mu},$$

*and the total $\ell_2$ error of the mechanism is bounded by $\mathbb{E}\left[\|\mathrm{Sk}_{0,\mu}(f(D)) - f(D)\|_2^2\right] \leq d\mu$.*

---

[3]We only consider the symmetric version of Skellam, but it is often more generally defined as the difference of independent Poisson random variables with different variances.

The Skellam mechanism was first introduced in [49] for the scalar case. As our goal is to apply the Skellam mechanism in the learning context, we have to address the following challenges. (1) *Tight privacy compositions:* Learning algorithms are iterative in nature and require the application of the DP mechanism many times (often $> 1000$). The current direct approximate DP analysis in [49] can be combined with advanced composition (AC) theorems [28, 22] but that leads to poor privacy-accuracy trade-offs (see Fig. 1). (2) *Privacy analysis for multi-dimensional queries:* In learning algorithms, the differentially private queries are multi-dimensional (where the dimension equals the number of model parameters, typically $\geq 10^6$). Using composition theorems lead to poor accuracy-privacy trade-offs and a direct extension of approximate DP guarantee [49] for the multi-dimensional case leads to a strong dependence on $\ell_1$ sensitivity which is prohibitively large in high dimensions. (3) *Data discretization:* The gradients are naturally continuous vectors but we would like to apply an integer based mechanism. This requires properly discretizing the data while making sure that the norm of the vectors (sensitivity of the query) is preserved. We will tackle challenges (1) and (2) in the remainder of this section and leave (3) for the next section.

## 3.1 Tight Numerical Accounting via Privacy Loss Distributions

We begin by defining the notion of privacy loss distributions (PLDs).

**Definition 3.3** (Privacy Loss Distribution). *For a multi-dimensional discrete privacy mechanism $M$ and neighboring datasets $D, D'$, for any $x \in \mathbb{Z}^d$, we define $f(x) = \log\left(\frac{P(M(D)=x)}{P(M(D')=x)}\right)$. The privacy loss random variable of $M$ at $(D, D')$ is $Z_{D,D'} = f(M(D))$ [22]. The privacy loss distribution (PLD) of $M$, denoted by $\mathrm{PLD}_{D,D'}$, is the distribution of $Z_{D,D'}$.*

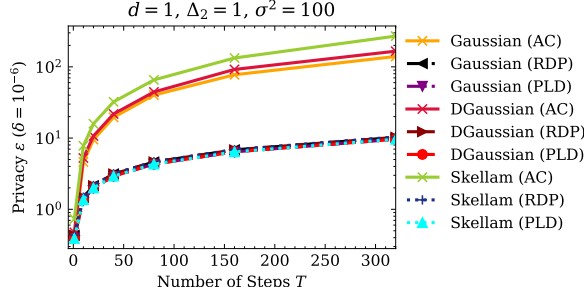

$d = 1, \Delta_2 = 1, \sigma^2 = 100$

Figure 1: Comparing privacy compositions across various mechanisms and accounting methods.

The PLD of a mechanism $M$ can be used to characterize its $(\varepsilon, \delta)$-DP guarantees.

**Lemma 3.4.** *A mechanism $M$ is $(\varepsilon, \delta)$-DP if and only if $\delta \geq \mathbb{E}_{Z \sim \mathrm{PLD}_{D,D'}}\left[1 - e^{\varepsilon - Z}\right]_+$ for all neighboring datasets $D, D'$ where $[x]_+ = \max(0, x)$.*

When a mechanism $M$ is applied $T$ times on a dataset, the overall PLD of the composed mechanism at $(D, D')$ is the $T$-fold convolution of $\mathrm{PLD}_{D,D'}$ [22]. Since discrete convolutions can be computed efficiently using fast Fourier transforms (FFTs) and the expectation in Lemma 3.4 can be numerically approximated, PLDs are attractive for tight numerical accounting [30, 34, 17]. Applying the above to the Skellam mechanism, a direct calculation shows that with $X_i$ are i.i.d. according to $\mathrm{Sk}_{0,\mu}$,

$$Z_{D,D'} = \sum_{i=1}^{d} \log\left(\frac{I_{X_i - f(D)_i}(\mu)}{I_{X_i - f(D')_i}(\mu)}\right).$$

When $d = 1$, it suffices to look at $Z = \log(I_{X - \Delta}(\mu)/I_X(\mu))$, where $\Delta = \max_{D,D'} |f(D) - f(D')|$ and $X \sim \mathrm{Sk}_{0,\mu}$. Since $X$ has a discrete and symmetric probability distribution and the $\log$ function is monotonic, the distribution of $Z$ can be easily characterized. This gives us a tight numerical accountant for the Skellam mechanism in the scalar case, which we use to compare it with both the Gaussian and discrete Gaussian mechanisms. Fig. 1 shows this comparison, highlighting the competitiveness of the Skellam mechanism and the problem of combining the direct analysis of [49] with advanced composition (AC) theorems. When $d > 1$, there are combinatorially many $Z_{D,D'}$'s that need to be considered, even when the $\ell_2$ sensitivity of $f(D)$ is bounded. The discrete Gaussian mechanism faces a similar issue (see Theorem 15 of [15]). To provide a tight privacy analysis in the multi-dimensional case, we prove a bound on the RDP guarantees of the Skellam mechanism in the next subsection. Fig. 1 and 2 show that our bound is tight and the competitiveness of the Skellam mechanism in high dimensions.

## 3.2 Tight Accounting via Rényi Differential Privacy

The following theorem states our main theoretical result, providing a relatively sharp bound on the RDP properties for the Skellam machanism.

**Theorem 3.5.** *For $\alpha \in \mathbb{Z}$, $\alpha > 1$ and sensitivity $\Delta \in \mathbb{Z}$, the Skellam Mechanism is $(\alpha, \varepsilon)$-RDP with*

$$\varepsilon(\alpha) \leq \frac{\alpha \Delta^2}{2\mu} + \min\left(\frac{(2\alpha - 1)\Delta^2 + 6\Delta}{4\mu^2}, \frac{3\Delta}{2\mu}\right), \tag{3.1}$$

To remind the reader in comparison, the Gaussian mechanism is $(\alpha, \varepsilon)$-RDP with $\varepsilon(\alpha) = \frac{\alpha\Delta^2}{2\mu}$. The bound we provide is at most $1 + O(1/\mu)$ worse than the bound for the Gaussian, which is negligible for all practical choices of $\mu$, especially as the privacy requirements increase.[4] Next we show a simple corollary which follows via the independent composition of RDP across dimensions.

**Corollary 3.6.** *The multi-dimensional Skellam Mechanism is $(\alpha, \varepsilon)$-RDP with*

$$\varepsilon(\alpha) \leq \frac{\alpha \Delta_2^2}{2\mu} + \min\left(\frac{(2\alpha - 1)\Delta_2^2 + 6\Delta_1}{4\mu^2}, \frac{3\Delta_1}{2\mu}\right). \tag{3.2}$$

*where $\Delta_1$ and $\Delta_2$ are the $\ell_1$ and $\ell_2$ sensitivities respectively.*

### 3.2.1 Proof Overview for Theorem 3.5

In this subsection, we provide the proof of Theorem 3.5 assuming a technical bound on the ratios of Bessel functions presented as Lemma 3.7, which is the core of our analysis and may be of independent interest. We provide a proof overview for Lemma 3.7, deferring the full proof to the appendix.

On a macroscopic level, our proof structure mimics the RDP proof for the Gaussian mechanism [36], and the main object of our interest is to bound the following quantity, defined for any $X, \Delta, \alpha$:

$$\Phi_{X,\alpha,\Delta}(\mu) \triangleq \log\left(\frac{I_{X-\Delta}(\mu)}{I_{X-\alpha\Delta}(\mu)}\left(\frac{I_{X-\Delta}(\mu)}{I_X(\mu)}\right)^{\alpha-1}\right). \tag{3.3}$$

The following lemma states our main bound on this quantity.

**Lemma 3.7.** *For any $X, \alpha \in \mathbb{N}$, with $\alpha > 1$ and $\Delta \in \mathbb{Z}$, we have that for all $\mu \geq 0$*

$$\Phi_{X,\alpha,\Delta}(\mu) \leq \frac{\alpha(\alpha-1)\Delta^2}{2\mu} + \min\left(\frac{(2\alpha-1)(\alpha-1)\Delta^2}{4\mu^2} + \frac{3(\alpha-1)|\Delta|}{2\mu^2}, \frac{3(\alpha-1)|\Delta|}{2\mu}\right).$$

Note that in contrast if we consider the analogous notion of $\Phi$ for the Gaussian mechanism (replacing $I_X(\mu)$ with the Gaussian density $e^{-X^2/2\mu}$), we readily get the bound $\frac{\alpha(\alpha-1)\Delta^2}{2\mu}$, which is the same as our bound up to lower order terms. We now provide the proof of Theorem 3.5.

*Proof of Theorem 3.5.* By RDP definition (2.2), we need to bound the following for any $\Delta, \alpha \geq 1$,

$$D_\alpha\left(\mathrm{Sk}_{\Delta,\mu}, \mathrm{Sk}_{0,\mu}\right) = \frac{1}{\alpha-1}\log\left(\sum_{X=-\infty}^{\infty} e^{-\mu} I_{X-\Delta}(\mu)\left(\frac{I_{X-\Delta}(\mu)}{I_X(\mu)}\right)^{\alpha-1}\right)$$

Now consider the following calculations on the log term:

$$\log\left(\sum_{X=-\infty}^{\infty} \frac{I_{X-\Delta}(\mu)}{e^\mu}\left(\frac{I_{X-\Delta}(\mu)}{I_X(\mu)}\right)^{\alpha-1}\right) = \log\left(\sum_{X=-\infty}^{\infty} \frac{I_{X-\alpha\Delta}(\mu)}{e^\mu} e^{\Phi_{X,\alpha,\Delta}(\mu)}\right)$$

$$\leq \log\left(\sum_{X=-\infty}^{\infty} e^{-\mu} I_{X-\alpha\Delta}(\mu)\right) + \max_{X\in\mathbb{Z}} \Phi_{X,\alpha,\Delta}(\mu)$$

$$\leq \frac{\alpha(\alpha-1)\Delta^2}{2\mu} + \min\left(\frac{(2\alpha-1)(\alpha-1)\Delta^2}{4\mu^2} + \frac{3(\alpha-1)|\Delta|}{2\mu^2}, \frac{3(\alpha-1)|\Delta|}{2\mu}\right),$$

where the inequality follows from Lemma 3.7. □

---

[4]The restriction that $\alpha$ needs to be an integer is a technical one owing to known bounds on Bessel functions. In practice as we show, this restriction has a negligible effect.

We now provide an overview for the proof of Lemma 3.7 highlighting the crux of the argument. As a first step we collect some known facts regarding Bessel functions. It is known that for $x \geq 0$ and $\nu \in \mathbb{Z}$, $\nu \geq 0$, $I_\nu(x)$ is a decreasing function in $\nu$, $I_{-\nu}(x) = I_\nu(x)$ and $\frac{I_{\nu-1}(\mu)}{I_\nu(x)}$ is an increasing function in $\nu$ [47]. A succession of works consider bounding the ratio of successive Bessel functions $I_{\nu-1}(x)/I_\nu(x)$, which is a natural quantity to considering the objective in Lemma 3.7. We use the following very tight characterization for this recently proved in [44, Theorem 5].

**Lemma 3.8.** *For any $\nu \geq 1/2, x \geq 0$ define the following function we have that*

$$\mathrm{arcsinh}(\delta_0(\nu, x)) \leq \log(I_{\nu-1}(x)) - \log(I_\nu(x)) \leq \mathrm{arcsinh}(\delta_2(\nu, x))$$

*where $\delta_\alpha(\nu, x)$ is defined as $\delta_\alpha(\nu, x) \triangleq \frac{\nu - 1/2}{x} + \frac{\nu + (\alpha-1)/2}{2x\sqrt{(\nu+(\alpha-1)/2)^2 + x^2}}$.*

Standard bounds such as those appearing in [5, 49] lead to the following conclusion:

$$\mathrm{arcsinh}\left((\nu - 1/2)/x\right) \leq \log(I_{\nu-1}(x)) - \log(I_\nu(x)) \leq \mathrm{arcsinh}\left(\nu/x\right).$$

While the above bound is significantly easier to work with, it leads to an RDP guarantee of Gaussian RDP + $O(\frac{\Delta}{\mu})$. In high dimensions this manifests as $O(\frac{\Delta_1}{\mu})$ and overall leads to a constant multiplicative factor over the Gaussian. On the other hand we prove a Gaussian RDP + $o_\mu(1)$ bound. Our proof of Lemma 3.7 splits into various cases depending on the signs of the quantities involved. We show the derivation for a single case below and defer the full proof to the appendix.

*Proof of Lemma 3.7 in the case $X \geq \alpha\Delta$, $\Delta \geq 0$.* Replacing $Y = X - \alpha\delta$ we get that

$$\Phi_{X,\alpha,\Delta}(\mu) = \log\left(\frac{I_{Y+(\alpha-1)\Delta}(\mu)}{I_Y(\mu)}\left(\frac{I_{Y+(\alpha-1)\Delta}(\mu)}{I_{Y+\alpha\Delta}(\mu)}\right)^{\alpha-1}\right)$$

$$= \sum_{j=0}^{\alpha-2}\left(\sum_{i=Y+j\Delta+1}^{Y+j\Delta+\Delta}\left(\log\left(\frac{I_{i-1+(\alpha-1-j)\Delta}(\mu)}{I_{i+(\alpha-1-j)\Delta}(\mu)}\right) - \log\left(\frac{I_{i-1}(\mu)}{I_i(\mu)}\right)\right)\right)$$

$$\leq \sum_{j=0}^{\alpha-2}\left(\sum_{i=Y+j\Delta+1}^{Y+j\Delta+\Delta}\left(\delta_2(i + (\alpha-1-j)\Delta, \mu) - \delta_0(i, \mu)\right)\right)$$

$$\leq \frac{\alpha(\alpha-1)\Delta^2}{2\mu} + \min\left(\frac{\alpha(\alpha-1)\Delta^2 + 2(\alpha-1)\Delta}{4\mu^2}, \frac{(\alpha-1)\Delta}{2\mu}\right),$$

where the first inequality follows from Lemma 3.8 and the fact that for all $0 \leq x \leq y$, $\mathrm{arcsinh}(y) - \mathrm{arcsinh}(x) \leq y - x$ and the second inequality follows from Lemma A.1 (provided in the appendix):

$$\delta_2(\nu_1, x) - \delta_0(\nu_2, x) \leq \frac{\nu_1 - \nu_2}{x} + \frac{1}{2x}\min\left(\frac{\nu_1 - \nu_2 + 1}{x}, 1\right).$$

$\square$

# 4 Applying the Skellam Mechanism to Federated Learning

With a sharp RDP analysis for the multi-dimensional Skellam mechanism presented in the previous section, we are now ready to apply it to differentially private federated learning. We first outline the general problem setting and then describe our approach under central and distributed DP models.

**Problem setting** At a high-level, we consider the distributed mean estimation problem. There are $n$ clients each holding a vector $x_i$ in $\mathbb{R}^d$ such that for all $i$, the vector norm is bounded as $\|x_i\|_2 \leq c$ for some $c \geq 0$. We denote the set of vectors as $\mathcal{X} = \{x_i\}_{i=1}^n$, and the aim is for each client to communicate the vectors $x_i$ to a central server which then aggregates them as $\widehat{x} = \frac{1}{n}\sum_i x_i$ for an external analyst. In federated learning, the client vectors $x_i$ are the model gradients or model deltas (typically $d \geq 10^6$) after training on the clients' local datasets, and this procedure can be repeated for many rounds ($T > 1000$). A large $d$ and $T$ thus necessitate accounting methods that provide tight privacy compositions for high-dimensional queries.

We are primarily concerned with three metrics for this procedure and their trade-offs: (1) *Privacy*: the mean $\widehat{x}$ should be differentially private with a reasonably small $(\varepsilon, \delta)$; (2) *Error*: we wish to

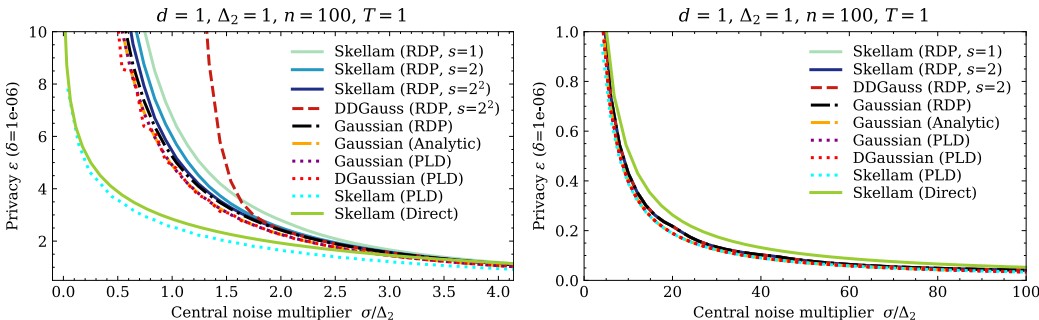

Figure 2: Benchmarking Skellam on sensitivity-1 queries under various accounting methods. RDP: Rényi DP. PLD: privacy loss distributions. Skellam (Direct): [49]. Gaussian (Analytic): [9]. DGaussian [15] / DDGauss [25]: central / distributed discrete Gaussian. $s$ is the scaling factor applied to both $\Delta$ and $\sigma$. For Skellam and DDGauss [25], the central noise with std $\sigma$ is split into $n$ shares each applied locally with std $\sigma/\sqrt{n}$; a large $n$ and small $\sigma$ can thus exacerbate the sum divergence term of DDGauss (left). **Left**: $\varepsilon \le 10$. **Right**: $\varepsilon \le 1$.

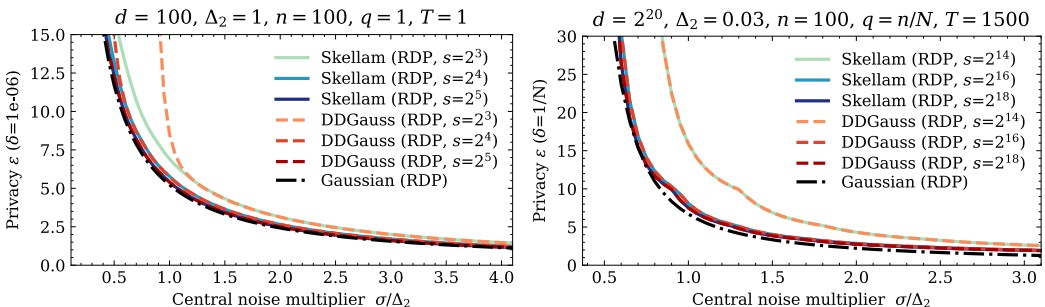

Figure 3: Comparing Skellam and Distributed Discrete Gaussian (DDGauss) on multi-dimensional real-valued queries, rounded to integers with $\beta = e^{-0.5}$ (Prop. 4.2). $s$ is the scaling applied to both $\sigma$ and $\Delta_2$; a larger $s$ reduces the rounding error and norm inflation. $q$ is the sampling rate. For Skellam and DDGauss [25], the central noise with std $\sigma$ is split into $n$ shares each applied locally with std $\sigma/\sqrt{n}$; a large $n$ and small $\sigma$ can exacerbate the sum divergence term of DDGauss. **Left:** Simple setting with $\Delta_2 = 1$. **Right:** FL-like setting for training CNNs on Federated EMNIST.

minimize the expected $\ell_2$ error; and (3) *Communication*: we wish to minimize the average number of bits communicated per coordinate. Characterizing this trade-off is an important research problem. For example, it has been recently shown [50] that without formal privacy guarantees, the client training data could still be revealed by the model updates $x_i$; on the other hand, applying differential privacy [48] to these updates can degrade the final utility.

**Skellam for central DP**    The central DP model refers to adding Skellam noise onto the non-private aggregate $\widehat{x}$ before releasing it to the external analyst. One important consideration is that the model updates in FL are continuous in nature, while Skellam is a discrete probability distribution. One approach is to appropriately discretize the client updates, e.g., via uniform quantization (which involves scaling the inputs by a factor $s \sim 2^b$ for some bit-width $b$ followed by stochastic rounding[5] for unbiased estimates), and the server can convert the private aggregate back to real numbers at the end. Note that this allows us to re-parameterize the variance of the added Skellam noise as $s^2\mu$, giving the following simple corollary based on Cor. 3.6:

**Corollary 4.1** (Scaled Skellam Mechanism). *With a scaling factor $s \in \mathbb{R}$, the multi-dimensional Skellam Mechanism is $(\alpha, \varepsilon)$-RDP with*

$$\varepsilon(\alpha) \le \frac{\alpha\Delta_2^2}{2\mu} + \min\left(\frac{(2\alpha-1)\Delta_2^2}{4s^2\mu^2} + \frac{3\Delta_1}{2s^3\mu^2}, \frac{3\Delta_1}{2s\mu}\right). \qquad (4.1)$$

---

[5]Example of stochastic rounding: 42.3 has 0.7 and 0.3 probability to be rounded to 42 and 43, respectively. Other discretization schemes are possible; we do not explore this direction further in this work.

As $s$ increases, the RDP of scaled Skellam rapidly approaches that of Gaussian as the second term above approaches 0, suggesting that under practical regimes with moderate compression bit-width, Skellam should perform competitively compared to Gaussian. Another aspect worth noting is that rounding vector coordinates from reals to integers can inflate the $\ell_2$-sensitivity $\Delta_2$, and thus more noise is required for the same privacy. To this end, we leverage the *conditional rounding* procedure introduced in [25] to obtain a bounded norm on the scaled and rounded client vector:

**Proposition 4.2** (Norm of stochastically rounded vector [25]). *Let $\tilde{x}$ be a stochastic rounding of vector $x \in \mathbb{R}^d$ to the integer grid $\mathbb{Z}^d$. Then, for $\beta \in (0,1)$, we have*

$$\mathbb{P}\left[\|\tilde{x}\|_2^2 \leq \|x\|_2^2 + d/4 + \sqrt{2\log(1/\beta)} \cdot \left(\|x\|_2 + \sqrt{d}/2\right)\right] \geq 1 - \beta. \tag{4.2}$$

Conditional rounding is thus defined as retrying the stochastic rounding on $x_i$ until $\|\tilde{x}_i\|_2^2$ is within the probabilistic bound above (which also gives the inflated sensitivity $\tilde{\Delta}_2$). We can then add Skellam noise to the aggregate $\sum_i \tilde{x}_i$ according to $\tilde{\Delta}_2$ before undoing the quantization (unscaling). Note that a larger scaling $s$ before rounding reduces the norm inflation and the extra noise needed (Fig. 3 right).

**Skellam for distributed DP with secure aggregation**    A stronger notion of privacy in FL can be obtained via the distributed DP model [25] that leverages secure aggregation (SecAgg [12]). The fact that the Skellam distribution is closed under summation allows us to easily extend from central DP to distributed DP. Under this model, the client vectors are quantized as in central DP model, but the Skellam noise is now added locally with variance $\mu/n$. Then, the noisy client updates are summed via SecAgg ($b$ bits per coordinate for field size $2^b$) which only reveals the noisy aggregate to the server. While the local noise might be insufficient for local DP guarantees, the aggregated noise at the server provides privacy and utility comparable to the central DP model, thus removing trust away from the central aggregator. Note that the modulo operations introduced by SecAgg does not impact privacy as it can be viewed as a post-processing of an already differentially private query.

We remark on several properties of the distributed Skellam compared to the distributed discrete Gaussian (DDGauss [25]). (1) DDGauss is not closed under summation, and the divergence between discrete Gaussians can lead to notable privacy degradation in settings such as quantile estimation [6] and federated analytics [42] with sufficiently large number of clients and small local noises (see also the left side of Fig. 2 and Fig. 3). While scaling mitigates this issue, it also requires additional bit-width which makes Skellam attractive under tight communication constraints. (2) Sampling from Skellam only requires sampling from Poisson, for which efficient implementations are widely available in numerical software packages. While efficient discrete Gaussian sampling has also been explored in the lattice-based cryptography community (e.g., [43, 18, 38]), we believe the accessibility of Skellam samplers would help facilitate the deployment of DP to FL settings with mobile and edge devices. See Appendix D for more discussion. (3) In practice where $s \gg 1$ (dictated by bit-width $b$), both Skellam (cf. Cor. 4.1) and DDGauss (with an exponentially small divergence) quickly approaches Gaussian under RDP, and any differences will be negligible (Fig. 3).

## 5   Empirical Evaluation

In this section, we empirically evaluate the Skellam mechanism on two sets of experiments: distributed mean estimation and federated learning. In both cases, we focus on the distributed DP model, but note that the Skellam mechanism can be easily adapted to the central DP setting as discussed in the earlier section. Unless otherwise stated, we use RDP accounting for all experiments due to the high-dimensional data and the ease of composition (Section 3). To obtain $\Delta_1$ for Skellam RDP, we note that $\Delta_1 \leq \Delta_2 \cdot \min(\sqrt{d}, \Delta_2)$ since $\Delta_1 \leq \sqrt{d}\Delta_2$ in general and $\Delta_1 \leq \Delta_2^2$ for integers.

Under the distributed DP model, we also introduce a random orthogonal transformation [29, 2, 25] before discretizing and aggregating the client vectors (which can be reverted after the aggregation); this makes the vector coordinates sub-Gaussian and helps spread the magnitudes of the vector coordinates across all dimensions, thus reducing the errors from quantization and potential wrap-around from SecAgg modulo operations. Moreover, by approximating the sub-exponential tail of the Skellam distribution as sub-Gaussian, we can derive a heuristic for choosing $s$ following [25] based on a bound on the variance $\tilde{\sigma}^2$ of the aggregated signal, as $\tilde{\sigma}^2 \leq c^2 n^2/d + n/(4s^2) + \mu$. We choose $s$ such that $2k\tilde{\sigma}$ are bounded within the SecAgg field size $2^b$, where $k$ is a small constant.

Algorithm 1 summarizes the aggregation procedure for the distributed Skellam mechanism via secure

---
**Algorithm 1** Aggregation Procedure for the Distributed Skellam Mechanism

---

**Inputs:** Private vector $x_i \in \mathbb{R}^{\bar{d}}$ for each client $i$; $\ell_2$ clip norm $c > 0$; Bit-width $b$; Target central noise variance $\mu > 0$; Number of clients $n$; Signal bound multiplier $k > 0$; Bias $\beta \in [0, 1)$.

**Shared randomness:** $d \times d$ diagonal matrix $D$ with uniformly random $\{-1, +1\}$ values, where $d \geq \bar{d}$ is the nearest power of 2.

**Shared scale:** Obtain scaling factor $s$ such that $2^b = 2k\tilde{\sigma} = 2k\sqrt{c^2n^2/d + n/(4s^2) + \mu}$.

**Procedure** CLIENTPROCEDURE$(x_i, s, D)$

  Clip and scale vector $\hat{x}_i = s \cdot \min(1, c/\|x_i\|_2) \cdot x_i$, and pad to $\tilde{d}$ dimensions with zeros.

  Random rotation: $\check{x}_i = \tilde{H}_d D \hat{x}_i$ where $\tilde{H}_d = \frac{1}{\sqrt{d}} H_d$ is the normalized $d \times d$ Hadamard matrix.

  **repeat** {conditional stochastic rounding}

    Stochastically round the coordinates of of $\check{x}_i$ to the integer grid to produce $\tilde{x}_i$

  **until** $\|\tilde{x}_i\|_2^2 \leq \min\left\{ \left(sc + \sqrt{d}\right)^2, s^2c^2 + d/4 + \sqrt{2\log(1/\beta)} \cdot \left(sc + \sqrt{d}/2\right)\right\}$.

  Local noising: Sample noise vector $y_i \in \mathbb{Z}^d$ where each entry is sampled from $\mathrm{Sk}_{0,s^2\mu/n}$.

  **return** $z_i = \tilde{x}_i + y_i$ under the SecAgg protocol with modulo bit-width $b$.

**Procedure** SERVERPROCEDURE$(z, s, D)$ {$z$ is the modular sum of $z_i$ under bit-width $b$}

  **return** $\bar{x} = \frac{1}{s}D\tilde{H}_d^\top z$, with $\bar{x} \approx \sum_i x_i \in \mathbb{R}^d$.

---

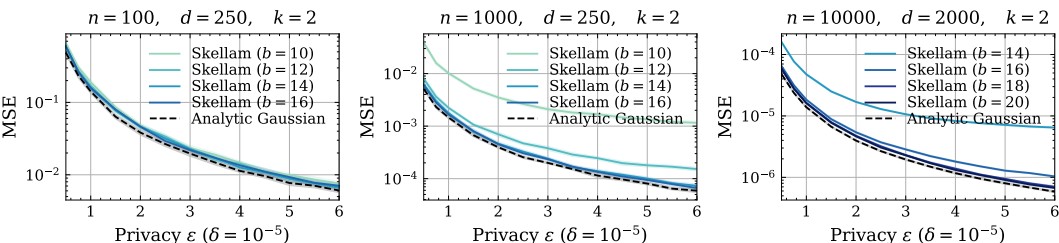

Figure 4: Distributed mean estimation with the distributed Skellam mechanism.

aggregation as well as the parameters used for the experiments. In summary, we have an $\ell_2$ clip norm $c > 0$; per-coordinate bit-width $b$; target central noise variance $\mu > 0$; number of clients $n$; signal bound multiplier $k > 0$; and rounding bias $\beta \in [0, 1)$. We fix $\beta = e^{-1/2}$ for all experiments. Note that the per-coordinate bit-width $b$ is for the aggregated sum as it determines the field size of SecAgg. For federated learning, we also consider the number of rounds $T$ and the total number of clients $N$ (thus the uniform sampling ratio $q = n/N$ at every round). Our experiments are implemented in Python, TensorFlow Privacy [32], and TensorFlow Federated [24]. See also Appendix for additional results and more details on the experimental setup.

## 5.1 Distributed Mean Estimation (DME)

We first consider DME as the generalization of (single round) FL. We randomly generate $n$ client vectors $X = \{x_i\}_{i=1}^n$ from the $d$-dimensional $\ell_2$ sphere with radius $c = 10$, and compute the true mean $\hat{x} = \frac{1}{n}\sum_i^n x_i$. We then compute the private estimate of $\hat{x}$ with the distributed Skellam mechanism (Algorithm 1) as $\bar{x}$. For a strong baseline, we use the analytic Gaussian mechanism [9] with tight accounting (see also Figure 2). In Figure 4, we plot the MSE as $\|\hat{x} - \bar{x}\|_2^2/d$ with 95% confidence interval (small shaded region) over 10 dataset initializations across different values of $b$, $d$, and $n$. Results demonstrate that Skellam can match Gaussian even with $n = 10000$ clients as long as the bit-width is sufficient. We emphasize that the communication cost $b$ depends logarithmically on $n$, and to put numbers into context, Google's production next-word prediction models [23, 39] use $n \leq 500$ and the production DP language model [40] uses $n = 20000$.

## 5.2 Federated Learning

**Setup** We evaluate on three public federated datasets with real-world characteristics: Federated EM-NIST [16], Shakespeare [31, 14], and Stack Overflow next word prediction (SO-NWP [8]). EMNIST is an image classification dataset for hand-written digits and letters; Shakespeare is a text dataset for next-character-prediction based on the works of William Shakespeare; and SO-NWP is a large-scale

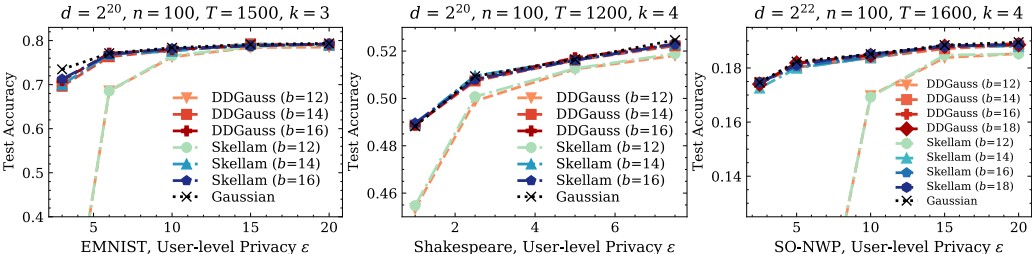

Figure 5: Federated learning with the distributed Skellam mechanism. DDGauss: Distributed Discrete Gaussian [25]. **Left / Middle / Right**: Test accuracies on EMNIST / Shakespeare / Stack Overflow NWP across different $\varepsilon$ and $b$. $\delta$ is set to $1/N$, $10^{-6}$, $10^{-6}$, respectively. For Shakespeare, privacy is reported with a hypothetical population size $N = 10^6$.

text dataset for next-word-prediction based on user questions/answers from stackoverflow.com. We emphasize that all datasets have natural client heterogeneity that are representative of practical FL problems: the images in EMNIST are grouped the writer of the handwritten digits, the lines in Shakespeare are grouped by the speaking role, and the sentences in SO-NWP are grouped by the corresponding Stack Overflow user. We train a small CNN with model size $\bar{d} < 2^{20}$ for EMNIST and use the recurrent models defined in [41] for Shakespeare and SO-NWP. The hyperparameters for the experiments follow those from [25, 6, 27, 41] and tuning is limited. For EMNIST, we follow [25] and fix $c = 0.03$, $n = 100$, $T = 1500$, client learning rate $\eta_{\text{client}} = 0.32$, server learning rate $\eta_{\text{server}} = 1$, and client batch size $m = 20$. For Shakespeare, we follow [6] and fix $n = 100$, $T = 1200$, $\eta_{\text{client}} = 1$, $\eta_{\text{server}} = 0.32$, and $m = 4$, and we sweep $c \in \{0.25, 0.5\}$. For SO-NWP, we follow [27] and fix $c = 0.3$, $n = 100$, $T = 1600$, $\eta_{\text{client}} = 0.5$, and $m = 16$, and we sweep $\eta_{\text{server}} \in \{0.3, 1\}$ and limit max examples per client to 256. In all cases, clients train for 1 epoch on their local datasets, and the client updates are weighted uniformly (as opposed to weighting by number of examples). See Appendix for more results and full details on datasets, models, and hyperparameters.

**Results** Figure 5 summarizes the FL experiments. For EMNIST and Shakespeare, we report the average test accuracy over the last 100 rounds. For SO-NWP, we report the top-1 accuracy (without padding, out-of-vocab, or begining/end-of-sentence tokens) on the test set. The results indicate that Skellam performs as good as Gaussian despite relying on generic RDP amplification via sampling [51] (cf. Fig. 3) and that Skellam matches DDG consistently under realistic regimes. This bears significant practical relevance given the advantages of Skellam over DDG in real-world deployments.

## 6 Conclusion

We have introduced the multi-dimensional Skellam mechanism for federated learning. We analyzed the Skellam mechanism through the lens of approximate DP, privacy loss distributions, and Rényi divergences, and derived a sharp RDP bound that enables Skellam to match Gaussian and discrete Gaussian in practical settings as demonstrated by our large-scale experiments. Since Skellam is closed under summation and efficient samplers are widely available, it represents an attractive alternative to distributed discrete Gaussian as it easily extends from the central DP model to the distributed DP model. Being a discrete mechanism can also bring potential communication savings over continuous mechanisms and make Skellam less prone to attacks that exploit floating-point arithmetic on digital computers. Some interesting future work includes: (1) our scalar PLD analysis for Skellam suggests room for improvements on our multi-dimensional analysis via a complete PLD characterization, and (2) our results on FL may be further improved via a targeted analysis for RDP amplification via sampling akin to [37]. Overall, this work is situated within the active area of private machine learning and aims at making ML more trustworthy. One potential negative impact is that our method could be (deliberately or inadvertently) misused, such as sampling the wrong noise or using a minuscule scaling factor, to provide non-existent privacy guarantees for real users' data. We nevertheless believe our results have positive impact as they facilitate the deployment of differential privacy in practice.

## Funding Transparency Statement

The authors were employed at and directly supported by Google. No third party funding was received by any of the authors to pursue this work.

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
