# A  Proof of Lemma 3.7

Before moving forward we state the following lemma which follows via a simple calculation.

**Lemma A.1.** *For any positive real number $\nu \geq 1/2$ and any $x \geq 0$, we have that*

$$\delta_0(\nu, x) \geq \frac{\nu - 1/2}{x}$$

$$\delta_2(\nu, x) \leq \min\left(\frac{\nu - 1/2}{x}\left(1 + \frac{1}{2x}\right) + \frac{1}{2x^2}, \frac{\nu - 1/2}{x} + \frac{1}{2x}\right).$$

*Further for any positive reals $\nu_1 \geq \nu_2, x$ we have that*

$$\delta_2(\nu_1, x) - \delta_0(\nu_2, x) \leq \frac{\nu_1 - \nu_2}{x} + \frac{1}{2x}\min\left(\frac{\nu_1 - \nu_2 + 1}{x}, 1\right)$$

*where $\delta_\alpha(\nu, x) \triangleq \frac{\nu - 1/2}{x} + \frac{\nu + (\alpha - 1)/2}{2x\sqrt{(\nu + (\alpha - 1)/2)^2 + x^2}}$ as defined in Lemma 3.8.*

*Proof.* The first inequality follows easily from the definition of $\delta$ and by noting that the function $\frac{x}{\sqrt{1+x^2}} \leq \min(x, 1)$. For the second inequality, by the definition of $\delta$ we have that

$$\delta_2(\nu_1, x) - \delta_0(\nu_2, x) = \frac{\nu_1 - \nu_2}{x} + \frac{1}{2x}\left(\frac{\nu_1 + 1/2}{\sqrt{(\nu_1 + 1/2)^2 + x^2}} - \frac{\nu_2 - 1/2}{\sqrt{(\nu_2 - 1/2)^2 + x^2}}\right).$$

Now, consider the scalar function $f(x) = \frac{x}{\sqrt{1+x^2}}$ for $x \geq 0$. Note that the function is monotonically increasing, concave and has values between $[0, 1]$ with $f'(x) \leq f'(0) = 1$. Putting these facts together we have that for any $x_1 \geq x_2$

$$\frac{x_1}{\sqrt{1 + x_1^2}} - \frac{x_2}{\sqrt{1 + x_2^2}} \leq \min(x_1 - x_2, 1).$$

$\square$

Using Lemma 3.8 and Lemma A.1 we have the following Lemma

**Lemma A.2.** *Given two non-negative integers $\nu, \Delta$ we have that*

$$\log\left(\frac{I_\nu}{I_{\nu+\Delta}}\right) \leq \sum_{j=\nu+1}^{j=\nu+\Delta} \operatorname{arcsinh}(\delta_2(j, x)) \leq \min\left(\frac{\Delta^2 + 2\nu\Delta}{2x}\left(1 + \frac{1}{2x}\right) + \frac{\Delta}{2x^2}, \frac{\Delta^2 + 2\nu\Delta}{2x} + \frac{\Delta}{2x}\right)$$

$$\log\left(\frac{I_\nu(x)}{I_{\nu+\Delta}(x)}\right) \geq \sum_{j=\nu+1}^{j=\nu+\Delta} \operatorname{arcsinh}(\delta_0(j, x))$$

We are now ready to provide the proof of Lemma 3.7.

*Proof of Lemma 3.7.* We prove the statement for $\Delta \geq 0$, a similar analysis applies for the case $\Delta \leq 0$ by switching $X$ to $-X$.

Since 3.8 applies only in the case when $\nu$ is positive, we need to handle the negative case via noting that for integer $\nu$ $I_\nu(x) = I_{|\nu|}(x)$. This necessitates the requirement for multiple cases. We begin with the first case

**Case 1 - $X \geq \alpha\Delta$**

In this case replacing setting $Y = X - \alpha\delta$ we get that

$$\Phi_{X,\alpha,\Delta}(\mu) = \log\left(\frac{I_{Y+(\alpha-1)\Delta}(\mu)}{I_Y(\mu)}\left(\frac{I_{Y+(\alpha-1)\Delta}(\mu)}{I_{Y+\alpha\Delta}(\mu)}\right)^{\alpha-1}\right), \tag{A.1}$$

where we know that $Y \geq 0$. Now consider the following calculation.

$$\Phi_{X,\alpha,\Delta}(\mu) \triangleq (\alpha - 1) \left( \sum_{i=Y+(\alpha-1)\Delta+1}^{Y+\alpha\Delta} \log\left(\frac{I_{i-1}(\mu)}{I_i(\mu)}\right) \right) - \sum_{i=Y+1}^{Y+(\alpha-1)\Delta} \log\left(\frac{I_{i-1}(\mu)}{I_i(\mu)}\right)$$

$$= \sum_{j=0}^{\alpha-2} \left( \sum_{i=Y+(\alpha-1)\Delta+1}^{Y+\alpha\Delta} \log\left(\frac{I_{i-1}(\mu)}{I_i(\mu)}\right) - \sum_{i=Y+j\Delta+1}^{Y+j\Delta+\Delta} \log\left(\frac{I_{i-1}(\mu)}{I_i(\mu)}\right) \right)$$

$$= \sum_{j=0}^{\alpha-2} \left( \sum_{i=Y+j\Delta+1}^{Y+j\Delta+\Delta} \left( \log\left(\frac{I_{i-1+(\alpha-1-j)}(\mu)}{I_{i+(\alpha-1-j)\Delta}(\mu)}\right) - \log\left(\frac{I_{i-1}(\mu)}{I_i(\mu)}\right) \right) \right)$$

$$\leq \sum_{j=0}^{\alpha-2} \left( \sum_{i=Y+j\Delta+1}^{Y+j\Delta+\Delta} \left( \operatorname{arcsinh}(\delta_2(i+(\alpha-1-j)\Delta,\mu)) - \operatorname{arcsinh}(\delta_0(i,\mu)) \right) \right)$$

$$\leq \sum_{j=0}^{\alpha-2} \left( \sum_{i=Y+j\Delta+1}^{Y+j\Delta+\Delta} \left( \delta_2(i+(\alpha-1-j)\Delta,\mu) - \delta_0(i,\mu) \right) \right)$$

$$\leq \sum_{j=0}^{\alpha-2} \left( \frac{(\alpha-1-j)\Delta^2}{\mu} + \min\left( \frac{(\alpha-1-j)\Delta^2 + \Delta}{2\mu^2}, \frac{\Delta}{2\mu} \right) \right)$$

$$= \frac{\alpha(\alpha-1)\Delta^2}{2\mu} + \min\left( \frac{\alpha(\alpha-1)\Delta^2 + 2(\alpha-1)\Delta}{4\mu^2}, \frac{(\alpha-1)\Delta}{2\mu} \right),$$

where the first inequality follows from Lemma 3.8 and the second inequality follows from the fact that for all $0 \leq x \leq y$ we have that $\operatorname{arcsinh}(y) - \operatorname{arcsinh}(x) \leq y - x$ and the third inequality follows from Lemma A.1.

**Case 2 - $X \leq 0$**

In this case replacing setting $Y = -X$ we get that

$$\Phi_{X,\alpha,\Delta}(\mu) = \log\left( \frac{I_{Y+\Delta}(\mu)}{I_{Y+\alpha\Delta}(\mu)} \left( \frac{I_{Y+\Delta}(\mu)}{I_Y(\mu)} \right)^{\alpha-1} \right). \tag{A.2}$$

where we know that $Y \geq 0$. Now consider the following calculation.

$$\Phi_{X,\alpha,\Delta}(\mu) = \sum_{i=Y+\Delta+1}^{Y+\alpha\Delta} \log\left(\frac{I_{i-1}(\mu)}{I_i(\mu)}\right) - (\alpha-1)\left( \sum_{i=Y+1}^{Y+\Delta} \log\left(\frac{I_{i-1}(\mu)}{I_i(\mu)}\right) \right)$$

$$= \sum_{j=1}^{\alpha-1} \left( \sum_{i=Y+j\Delta+1}^{Y+j\Delta+\Delta} \log\left(\frac{I_{i-1}(\mu)}{I_i(\mu)}\right) - \sum_{i=Y+1}^{Y+\Delta} \log\left(\frac{I_{i-1}(\mu)}{I_i(\mu)}\right) \right)$$

$$= \sum_{j=1}^{\alpha-1} \left( \sum_{i=Y+1}^{Y+\Delta} \left( \log\left(\frac{I_{i-1+j\Delta}(\mu)}{I_{i+j\Delta}(\mu)}\right) - \log\left(\frac{I_{i-1}(\mu)}{I_i(\mu)}\right) \right) \right)$$

$$\leq \sum_{j=1}^{\alpha-1} \left( \sum_{i=Y+1}^{Y+\Delta} \left( \operatorname{arcsinh}(\delta_2(i+j\Delta,\mu)) - \operatorname{arcsinh}(\delta_0(i,\mu)) \right) \right)$$

$$\leq \sum_{j=1}^{\alpha-1} \left( \sum_{i=Y+1}^{Y+\Delta} \left( \delta_2(i+j\Delta,\mu) - \delta_0(i,\mu) \right) \right)$$

$$\leq \sum_{j=1}^{\alpha-1} \left( \frac{j\Delta^2}{\mu} + \min\left( \frac{j\Delta^2 + \Delta}{2\mu^2}, \frac{\Delta}{2\mu} \right) \right)$$

$$= \frac{\alpha(\alpha-1)\Delta^2}{2\mu} + \min\left( \frac{\alpha(\alpha-1)\Delta^2 + 2(\alpha-1)\Delta}{4\mu^2}, \frac{(\alpha-1)\Delta}{2\mu} \right),$$

where the first inequality follows from Lemma 3.8 and the second inequality follows from the fact that for all $0 \leq x \leq y$ we have that $\operatorname{arcsinh}(y) - \operatorname{arcsinh}(x) \leq y - x$ and the third inequality follows from Lemma A.1.

**Case 3 -** $X \in [0, \Delta/2]$

In this case we first note that

$$\Phi_{X,\alpha,\Delta}(\mu) = \log\left(\frac{I_{X-\Delta}(\mu)}{I_{X-\alpha\Delta}(\mu)}\left(\frac{I_{X-\Delta}(\mu)}{I_X(\mu)}\right)^{\alpha-1}\right)$$

$$= \log\left(\frac{I_{\Delta-X}(\mu)}{I_{\alpha\Delta-X}(\mu)}\left(\frac{I_{\Delta-X}(\mu)}{I_X(\mu)}\right)^{\alpha-1}\right)$$

Next consider the following calculation which corresponds to applying Lemma A.2 to the above expression we get that,

$$\Phi_{X,\alpha,\Delta}(\mu) \leq \sum_{j=\Delta-X+1}^{\alpha\Delta-X} \operatorname{arcsinh}(\delta_2(j,\mu)) - (\alpha-1)*\left(\sum_{j=X+1}^{\Delta-X} \operatorname{arcsinh}(\delta_0(j,\mu))\right)$$

We again intend to use the inequality $\operatorname{arcsinh}(y) - \operatorname{arcsinh}(x) \leq y - x$ for $y \geq x$. To this end first note that the number of terms on the left summation in the above inequality are at least as many as the number of terms on the RHS (taking the multiplicity via $\alpha - 1$ into account). Therefore for those terms we can apply the above inequality. For the remaining we simply use the inequality $\operatorname{arcsinh}(x) \leq x$. Therefore we get the following simplification,

$$\Phi_{X,\alpha,\Delta}(\mu) \leq \sum_{j=\Delta-X+1}^{\alpha\Delta-X} \delta_2(j,\mu) - (\alpha-1)*\left(\sum_{j=X+1}^{\Delta-X} \delta_0(j,\mu)\right).$$

We can now use the upper and lower bounds on $\delta_2, \delta_0$ given by Lemma A.1. To this end consider the following calculation,

$$\sum_{j=\Delta-X+1}^{\alpha\Delta-X} \frac{j-1/2}{\mu} - (\alpha-1)*\left(\sum_{j=X+1}^{\Delta-X} \frac{j-1/2}{\mu}\right)$$

$$= \frac{(\alpha-1)^2\Delta^2 + 2(\Delta-X)(\alpha-1)\Delta}{2} - (\alpha-1)\left(\frac{(\Delta-2X)^2 + 2X(\Delta-2X)}{2}\right)$$

$$= \frac{\alpha(\alpha-1)\Delta^2}{2\mu}.$$

Now a direct application of Lemma A.1 yields

$$\log\left(\frac{I_{\Delta-X}(\mu)}{I_{\alpha\Delta-X}(\mu)}\left(\frac{I_{\Delta-X}(\mu)}{I_X(\mu)}\right)^{\alpha-1}\right) \leq \frac{\alpha(\alpha-1)\Delta^2}{2\mu} + \min\left(\frac{(\alpha+1)(\alpha-1)\Delta^2}{4\mu^2} + \frac{(\alpha-1)\Delta}{2\mu^2}, \frac{(\alpha-1)\Delta}{2\mu}\right).$$

**Case 4 -** $X \in [\Delta/2, \Delta]$

In this case we first note that

$$\Phi_{X,\alpha,\Delta}(\mu) = \log\left(\frac{I_{X-\Delta}(\mu)}{I_{X-\alpha\Delta}(\mu)}\left(\frac{I_{X-\Delta}(\mu)}{I_X(\mu)}\right)^{\alpha-1}\right)$$

$$= \log\left(\frac{I_{\Delta-X}(\mu)}{I_{\alpha\Delta-X}(\mu)}\left(\frac{I_{\Delta-X}(\mu)}{I_X(\mu)}\right)^{\alpha-1}\right)$$

Next consider the following calculation which corresponds to applying Lemma A.2 to the above expression and collecting the terms corresponding to $\frac{\Delta^2+2\nu\Delta}{2x}$ in the lemma in this context.

$$\frac{(\alpha-1)^2\Delta^2 + 2(\Delta-X)(\alpha-1)\Delta}{2} + \frac{(2X-\Delta)^2 + 2(\Delta-X)(2X-\Delta)}{2} = \frac{\alpha(\alpha-1)\Delta^2}{2\mu}$$

Now a direct application of Lemma A.2 yields

$$\log\left(\frac{I_{\Delta-X}(\mu)}{I_{\alpha\Delta-X}(\mu)}\left(\frac{I_{\Delta-X}(\mu)}{I_X(\mu)}\right)^{\alpha-1}\right) \leq \frac{\alpha(\alpha-1)\Delta^2}{2\mu} + \min\left(\frac{\alpha(\alpha-1)\Delta^2}{4\mu^2} + \frac{3(\alpha-1)\Delta}{2\mu^2}, \frac{3(\alpha-1)\Delta}{2\mu}\right).$$

**Case 5 -** $X \in [\Delta, (\alpha+1)\Delta/2]$

In this case we first note that

$$\Phi_{X,\alpha,\Delta}(\mu) = \log\left(\frac{I_{X-\Delta}(\mu)}{I_{X-\alpha\Delta}(\mu)}\left(\frac{I_{X-\Delta}(\mu)}{I_X(\mu)}\right)^{\alpha-1}\right)$$

$$= \log\left(\frac{I_{X-\Delta}(\mu)}{I_{\alpha\Delta-X}(\mu)}\left(\frac{I_{X-\Delta}(\mu)}{I_X(\mu)}\right)^{\alpha-1}\right)$$

Next consider the following calculation which corresponds to applying Lemma A.2 to the above expression and collecting the terms corresponding to $\frac{\Delta^2+2\nu\Delta}{2x}$ in the lemma in this context.

$$\frac{(\alpha-1)(\Delta^2+2(X-\Delta)\Delta)}{2} + \frac{((\alpha+1)\Delta-2X)^2 + 2(X-\Delta)((\alpha+1)\Delta-2X)}{2} = \frac{\alpha(\alpha-1)\Delta^2}{2\mu}$$

Now a direct application of Lemma A.2 yields

$$\log\left(\frac{I_{X-\Delta}(\mu)}{I_{\alpha\Delta-X}(\mu)}\left(\frac{I_{X-\Delta}(\mu)}{I_X(\mu)}\right)^{\alpha-1}\right) \leq \frac{\alpha(\alpha-1)\Delta^2}{2\mu} + \min\left(\frac{\alpha(\alpha-1)\Delta^2}{4\mu^2} + \frac{(\alpha-1)\Delta}{2\mu^2}, \frac{(\alpha-1)\Delta}{2\mu}\right).$$

**Case 6 -** $X \in [(\alpha+1)\Delta/2, \alpha\Delta]$

In this case we first note that

$$\Phi_{X,\alpha,\Delta}(\mu) = \log\left(\frac{I_{X-\Delta}(\mu)}{I_{X-\alpha\Delta}(\mu)}\left(\frac{I_{X-\Delta}(\mu)}{I_X(\mu)}\right)^{\alpha-1}\right)$$

$$= \log\left(\frac{I_{X-\Delta}(\mu)}{I_{\alpha\Delta-X}(\mu)}\left(\frac{I_{X-\Delta}(\mu)}{I_X(\mu)}\right)^{\alpha-1}\right)$$

Next consider the following calculation which corresponds to applying Lemma A.2 to the above expression we get that,

$$\Phi_{X,\alpha,\Delta}(\mu) \leq (\alpha-1)*\left(\sum_{j=X-\Delta+1}^{X}\operatorname{arcsinh}(\delta_2(j,\mu))\right) - \sum_{j=\alpha\Delta-X+1}^{X-\Delta}\operatorname{arcsinh}(\delta_0(j,\mu))$$

We again intend to use the inequality $\operatorname{arcsinh}(y) - \operatorname{arcsinh}(x) \leq y - x$ for $y \geq x$. To this end first note that the number of terms on the left summation in the above inequality are at least as many as the number of terms on the RHS (taking the multiplicity via $\alpha-1$ into account). Therefore for those terms we can apply the above inequality. For the remaining we simply use the inequality $\operatorname{arcsinh}(x) \leq x$. Therefore we get the following simplification,

$$\Phi_{X,\alpha,\Delta}(\mu) \leq (\alpha-1)*\left(\sum_{j=X-\Delta+1}^{X}\delta_2(j,\mu)\right) - \sum_{j=\alpha\Delta-X+1}^{X-\Delta}\delta_0(j,\mu).$$

We can now use the upper and lower bounds on $\delta_2, \delta_0$ given by Lemma A.1. To this end consider the following calculation,

$$(\alpha-1)*\left(\sum_{j=X-\Delta+1}^{X}\frac{j-1/2}{\mu}\right) - \sum_{j=\alpha\Delta-X+1}^{X-\Delta}\frac{j-1/2}{\mu}$$

$$= \frac{(\alpha-1)(\Delta^2+2(X-\Delta)\Delta)}{2} - \frac{(2X-(\alpha+1)\Delta)^2 + 2(\alpha\Delta-X)((2X-(\alpha+1)\Delta)}{2}$$

$$= \frac{\alpha(\alpha-1)\Delta^2}{2\mu}.$$

Now a direct application of Lemma A.2 yields

$$\log\left(\frac{I_{X-\Delta}(\mu)}{I_{\alpha\Delta-X}(\mu)}\left(\frac{I_{X-\Delta}(\mu)}{I_X(\mu)}\right)^{\alpha-1}\right) \le \frac{\alpha(\alpha-1)\Delta^2}{2\mu} + \min\left(\frac{(2\alpha-1)(\alpha-1)\Delta^2}{4\mu^2} + \frac{(\alpha-1)\Delta}{\mu^2}, \frac{(\alpha-1)\Delta}{\mu}\right).$$

$\square$

## B Additional Results

### B.1 Shakespeare

We additionally evaluate our methods on Shakespeare, a public federated language modeling dataset [31, 41]. The dataset is built on the collective works of William Shakespeare, where each of the total $N = 715$ clients corresponds to a speaking role with at least two lines. The task of this dataset is to predict the next word based on the preceding words in a line. The dataset is split into training set and test set by partitioning the lines from each client. The model used for this task follows that of [41], to which we refer the reader for more details on the dataset and the experimental setup.

Our hyperparameters mostly follow those from [6, 41] and limited tuning was performed; see Table 2 for a complete list of hyperparameters. We do a small grid search over the $\ell_2$ clipping values $c \in \{0.25, 0.5\}$. We set the number of clients per round $n = 100$ and we train for $T = 1200$ rounds. In particular, note that it is challenging to obtain a small $\varepsilon$ on this dataset due to the small $N$ and the minimal $n$ sufficient for convergence; we thus follow [6] and [33] to report privacy with a hypothetical population size $N = 10^6$.

Figure 6 compares the test accuracies (averaged over the last 100 rounds) across different mechanisms and values of $\varepsilon$ and $b$. Figure 7 compares the mechanisms during training. The optimal clipping is $c = 0.5$ for $\varepsilon = 7.5$ and $c = 0.25$ otherwise. Note that Skellam matches DDGauss across all settings. The slight performance gap from both Skellam and DDGauss to Gaussian is likely due to the effects of modular clipping error from SecAgg (cf. Figure 10).

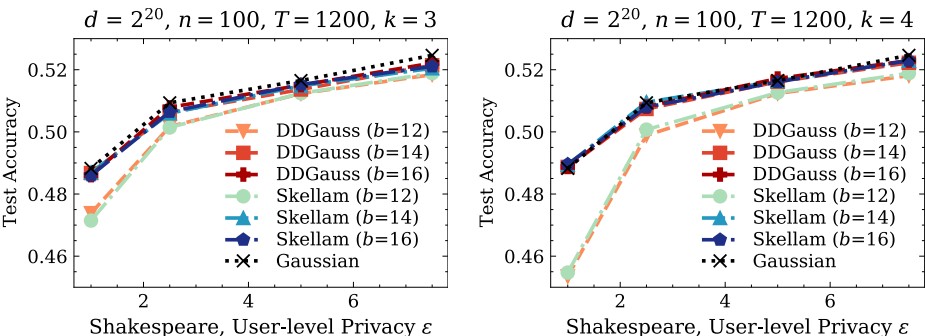

Figure 6: Summary of test accuracies on **Shakespeare** across different $\varepsilon$ and $b$ averaged over the last 100 rounds with a hypothetical population size $N = 10^6$. $\delta = 10^{-6}$. **Left:** $k = 3$. **Right:** $k = 4$.

### B.2 Federated EMNIST

Figure 8 includes the omitted details of Figure 5 (left) and illustrates the effect of a larger $k$. Figure 9 compares the test accuracies of different methods on EMNIST over training $T = 1500$ rounds for different values of $\varepsilon$ and $b$ with $k = 3$.

### B.3 Stack Overflow Next Word Prediction

Figure 10 illustrates the effect of $k$ on the test accuracies. Note that for $k = 3$, there is a slight performance gap between Gaussian and Skellam/DDGauss likely due to the modular clipping error introduced by SecAgg. By increasing $k$, we can reduce scaling and sacrifice quantization errors to close the gap. Figure 11 additionally shows the accuracies on the validation set over training $T = 1600$ rounds for different values of $\varepsilon$ and $b$ with $k = 4$.

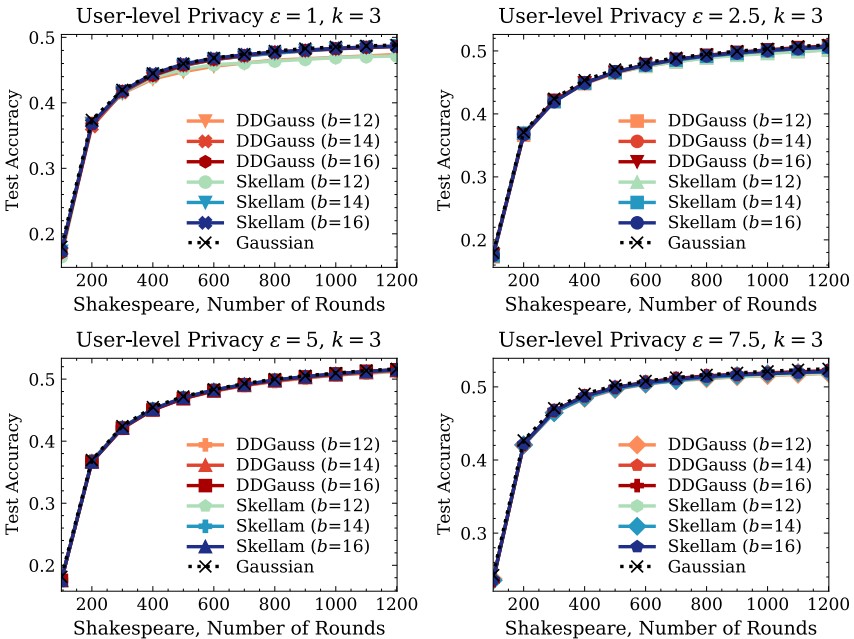

Figure 7: Test accuracies on **Shakespeare** over training $T = 1200$ rounds (averaged every 100 rounds) with a hypothetical population size $N = 10^6$. $\delta = 10^{-6}$. $k = 3$.

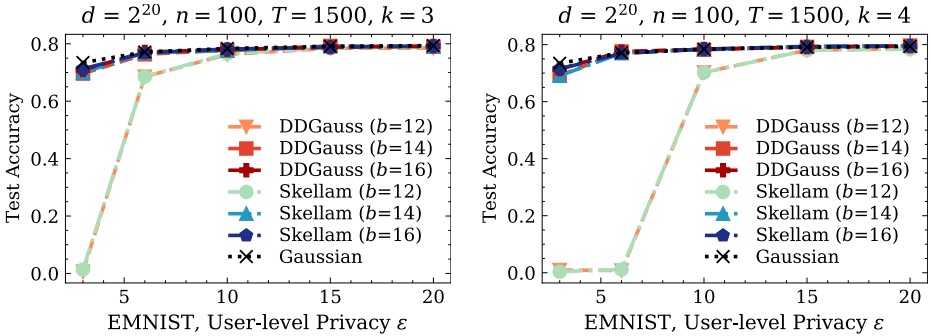

Figure 8: Summary of test accuracies on **Federated EMNIST** across different $\varepsilon$ and $b$ averaged over the last 100 rounds. $\delta = 1/N$. **Left:** $k = 3$. **Right:** $k = 4$.

### B.4 Comparison against Discrete Laplace

Figure 12 compares Skellam and (discrete) Gaussian against the discrete Laplace mechanism under various accounting schemes on privacy compositions.

## C Additional Details

### C.1 Models for FL experiments

For Federated EMNIST, we use a small convolutional network similar to the architecture used in [41]. Our architecture is slightly smaller and has $\bar{d} < 2^{20}$ parameters to reduce the zero padding required by the randomized Hadamard transform (see Algorithm 1). The architecture is summarized in Table 1. For Stack Overflow NWP and Shakespeare (Section B.1), we use the architectures from [41] directly.[6]

---

[6] https://github.com/google-research/federated/tree/master/utils/models

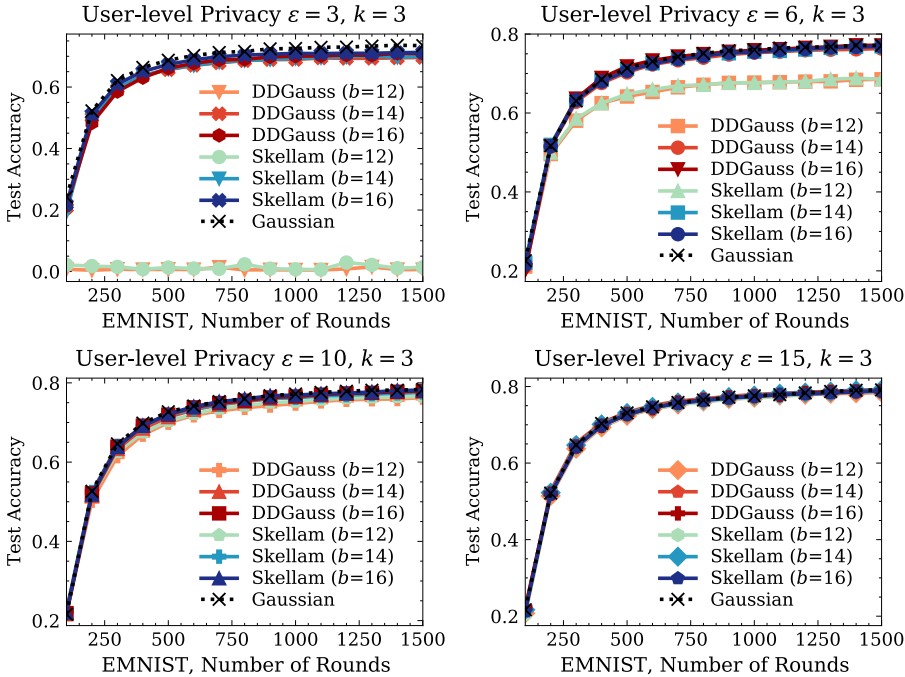

Figure 9: Test accuracies on **Federated EMNIST** over training $T = 1500$ rounds (averaged every 100 rounds). $\delta = 1/N$. $k = 3$.

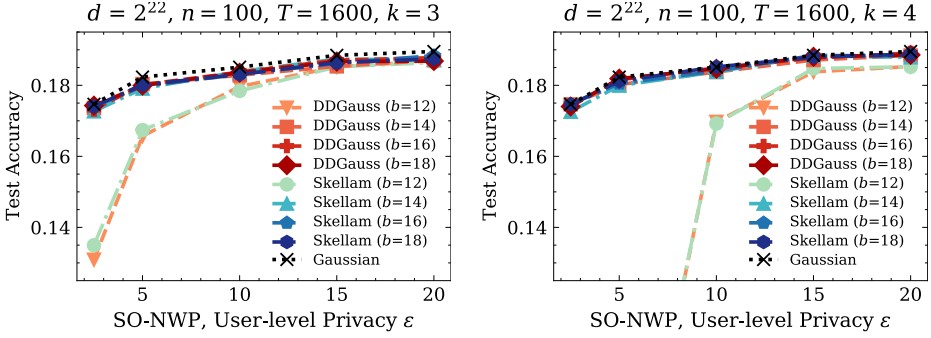

Figure 10: Summary of test accuracies on **Stack Overflow NWP** across different $\varepsilon$ and $b$. $\delta = 10^{-6}$. **Left:** $k = 3$. **Right:** $k = 4$.

## C.2 Datasets

**License, Availability, and Curation**   The federated datasets used in our FL experiments (Federated EMNIST [14], Stack Overflow Next Word Prediction [8], and Shakespeare [31]) are publicly available[7] from TensorFlow Federated [24]. To our knowledge, they have been appropriately anonymized and do not contain personally identifiable information. Federated EMNIST and Shakespeare is licensed under the BSD 2-Clause License, while SO-NWP uses the CC BY-SA 3.0 License.

**Dataset Splits**   For all FL datasets, we used the standard dataset split provided by TensorFlow. For EMNIST and Shakespeare, the datasets are split into training set and test set, and performance is reported on the test set. For Stack Overflow NWP, the dataset is split into training, validation, and test sets; the summary plots (e.g. Figure 10) report performance on the test set and the validation plots (e.g. Figure 11) report performance on the validation set. While other validation metrics/methods are possible, we note that using the dataset splits available from TensorFlow is standard practice in recent

---

[7]https://www.tensorflow.org/federated/api_docs/python/tff/simulation/datasets

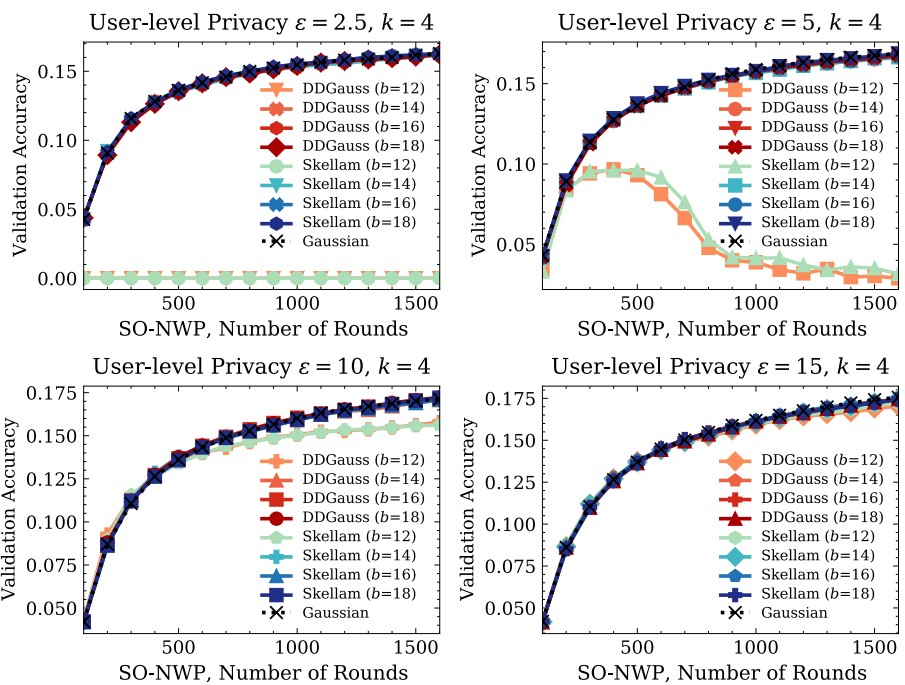

Figure 11: Validation accuracies on **Stack Overflow NWP** over training $T = 1600$ rounds (averaged every 100 rounds). $\delta = 10^{-6}$. $k = 4$.

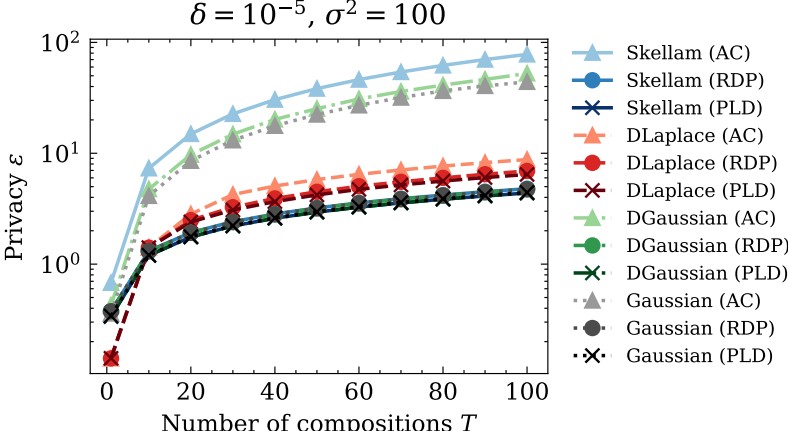

Figure 12: Comparing privacy compositions across various mechanisms in the scalar case. $\Delta = 1$. DLaplace: Discrete Laplace. DGaussian: Discrete Gaussian. AC: Advanced Composition. RDP: Rényi DP. PLD: Privacy Loss Distributions.

work (e.g. [41, 25, 33, 3]) and it allows our methods to be comparable in similar settings. Note also that validation data are often unavailable for training real-world FL models, and techniques such as $k$-fold validation can incur additional privacy costs.

### C.3 Other Implementation Details

**Privacy Amplification via Sub-sampling**    The privacy guarantees for FL experiments leverage amplification via sub-sampling as a subset of the clients is selected in each of the $T$ rounds to participate in training (we use [37] for Gaussian and [51] for Skellam and DDGauss). We note that the underlying assumptions – that the clients can be sampled uniformly, and that the identities of the clients can be hidden from the sampler – do not always hold, particularly in federated learning

| Layer | Output Shape | # Params |
|---|---|---|
| Input | $28 \times 28 \times 1$ | 0 |
| Conv $3 \times 3$, 32 | $26 \times 26 \times 32$ | 320 |
| MaxPool | $13 \times 13 \times 32$ | 0 |
| Conv $3 \times 3$, 64 | $11 \times 11 \times 64$ | 18496 |
| Dropout, 25% | $11 \times 11 \times 64$ | 0 |
| Flatten | 7744 | 0 |
| Dense | 128 | 991360 |
| Dropout, 25% | 128 | 0 |
| Dense | 62 | 7998 |
| Total # Params | | 1018174 |

Table 1: Model architecture for Federated EMNIST.

| | EMNIST | Shakespeare | Stack Overflow NWP |
|---|---|---|---|
| Client LR $\eta_{\text{client}}$ | 0.32 | 1 | 0.5 |
| Server LR $\eta_{\text{server}}$ | 1 | 0.32 | {0.3, 1} |
| Server momentum | 0.9 | 0.9 | 0.9 |
| Client Batch Size | 20 | 4 | 16 |
| Client epochs per round | 1 | 1 | 1 |
| Max examples per client | - | - | 256 |
| Clients uniform weighting | ✓ | ✓ | ✓ |
| $\ell_2$ clipping $c$ | 0.03 | {0.25, 0.5} | 0.3 |
| Clients per round $n$ | 100 | 100 | 100 |
| Population size $N$ | 3400 | 715‡ | 342477 |
| Training rounds $T$ | 1500 | 1200 | 1600 |
| Conditional rounding bias† $\beta$ | $\exp(-0.5)$ | $\exp(-0.5)$ | $\exp(-0.5)$ |
| Privacy $\delta$ | $1/N$ | $10^{-6}$ | $10^{-6}$ |

Table 2: Summary of hyperparameters for the FL experiments. †Discrete mechanisms only. ‡A hypothetical population size $N = 10^6$ was used when reporting privacy guarantees.

settings where the availability of the clients can be different at each round and the central server is the entity initiating the training/sampling procedure. However, we note that the privacy guarantees is applicable for external analysts that request the trained model from the central aggregator.

**Random seeds** For all experiments, we fixed both the seed for the client dataset randomness (sampling, shuffling, etc.) and the seed of initializing the parameters of the model architectures. Note that the FL experiments are not repeated for multiple seeds due to computational costs, though the test accuracies on EMNIST/Shakespeare and the validation accuracies on SO-NWP are averaged over the last 100 rounds.

**Hyperparameters** Table 2 summarizes the hyperparameters used for the FL experiments. We adopt most hyperparameters from previous work and tuning is limited. We follow [25] for EMNIST, [6] for Shakespeare, and [27] for Stack Overflow NWP. We sweep different server learning rates for SO-NWP and different clipping thresholds $c$ for Shakespeare and report the best results. We also show the effect of $k$ on the trade-off between modular clipping error and quantization error (e.g. Figure 10), though the same values of $k$ is used when comparing different methods.

# D    Practical Remarks

## D.1    Ease of Sampling

One of the practical advantages of the Skellam mechanism compared to the (distributed) discrete Gaussian mechanism is that the sampling routines for Skellam (Poisson) distribution are widely available in data analysis and machine learning software packages such as NumPy, TensorFlow, and PyTorch that ML / DP practitioners would use for development. However, we'd like to point out that

discrete Gaussian sampling itself has been well explored in the lattice-based cryptography community (e.g., [43, 18, 38]) and we would expect that the efficient sampling routines (often implemented in C/C++, e.g. [4]) would perform as good as optimized Poisson samplers if compared under a similar setting. While it is perceivable that discrete Gaussian samplers will become more accessible for ML and DP practitioners in the future, the discrete Gaussian has only recently been introduced [15] in the differential privacy context and thus the Skellam mechanism provides a practical advantage as differentially private FL systems with compression and scalable secure aggregation are near deployment today. We also provide a preliminary comparison of the sampling time against two existing implementations of discrete Gaussian sampler [15, 25] available to DP practitioners; the implementation from [15] is not vectorized and thus can be $1000\times$ slower than Skellam sampling, and the implementation from [25] is still up to 40% slower in TensorFlow eager execution.

## D.2 Closure Under Summation

The property that Skellam samples are closed under summation gives several practical advantages.

From a theoretical standpoint, being closed under summation removes the need for accounting for the divergence errors, which, as measured in DDGauss [25] as an infinity divergence, grow with the number of clients (as opposed to no such dependence for Skellam), which can be problematic with settings of massively distributed client base (e.g. federated analytics).

Concretely, consider the simple quantile estimation problem [6] where we have a large number of clients ($n \geq 1000$) and sensitivity-1 (binary) queries. With a large central noise standard deviation $\sigma_c \geq 5$, we can achieve a strong privacy (as $\varepsilon \approx \Delta/\sigma_c$), but if this noise is to be added locally (via the distributed DP with SecAgg model), the $\tau$ term of DDGauss (Thm 1 of [25]) would significantly degrade the privacy guarantees because large $n$ means more additive divergence terms and smaller local noise standard deviation which (exponentially) widens the divergence – this effect can be inferred from the left of Figure 2 where DDGauss underperforms at small noise levels and, in comparison, Skellam degrades more gracefully with smaller noises. For a specific example, consider $n = 10000$ and $\sigma_c = 50$; in this case, the local noise standard deviation $\sigma_l = 0.5$, and at $\alpha = 2$, the RDP $\varepsilon(\alpha)$ of Gaussian and Skellam is $4 \times 10^{-4}$ and $4.0036 \times 10^{-4}$ respectively, while the RDP of DDGauss is $> 723$ (a factor of $> 10^6$) due to the sum divergence term $\tau$. While scaling both the raw values and the noise variances can help alleviate this issue, it also introduces additional communication costs.

From an engineering perspective, while the divergence term for discrete Gaussian is usually small enough, we still need to keep track of it and its dependent parameters (number of clients, client dimensions, variance) for privacy accounting. Skellam on the other hand only requires us to track the variance and thereby allows easier switching between central DP (noise on the server) and distributed DP (noise on the clients) implementations.