# OpenReview forum: "The Skellam Mechanism for Differentially Private Federated Learning"
_NeurIPS.cc/2021/Conference — NeurIPS 2021 Poster_

### Official Review · Reviewer_p2AR · 2021-07-14

**Rating:** 7
**Confidence:** 3

**Summary:**

The paper introduces the multi-dimensional Skellam (difference of two Poisson random variables) mechanism for differential privacy. They show that the Skellam Mechanism is Renyi DP (RDP) with slightly worse guarantee than than Gaussian Mechanism. They also compare the privacy loss over several rounds against Gaussian and Discrete Gaussian using Renyi DP, Advanced Composition and the Privacy Loss Distribution. Results show that Skellam performs similarly to Discrete Gaussian and Gaussian when using RDP and PLD bounds.
As an application they consider using the Skellam Mechanism for distributed mean estimation and two Federated learning tasks (EMNIST and SO-NWP). Experiments show that they can match the performance of continuous Gaussian.

***
After rebuttal. I've read all the reviews and replies. Raising my score appropriately.

**Main Review:**

The paper is relatively easy to read. The technical claims seem to be correct and is novel in the sense that it introduces a mechanism that did not exist before. Unfortunately, I am not convinced this paper will really be impactful. One of the motivations for this work is that Discrete Gaussian is not part of Tensorflow, I’m almost certain this is just a matter of time. The main strength of this mechanism is that Skellam is closed under summation but DGaussian is not. It is unclear to me that this advantage is enough for publication of this paper. It is possible I missed something so please let me know if I did and I may revisit my score.

Another question, why does the PLD section exist if you have at your disposal Theorem 3.5?

Minor:
Line 130: I think the definition of f(x) is wrong. It should be P(M(D) = x), instead of M(D)=x (same with D’), right?
Line 106, typo. Z should be X.


**Time Spent Reviewing:**

4

---

> ### Author Response · Authors · 2021-08-10
> **Author response to initial review**
>
> We thank the reviewer for the comments.
>
> > “One of the motivations for this work is that Discrete Gaussian is not part of Tensorflow, I’m almost certain this is just a matter of time."
>
> The system for distributed DP via SecAgg is near deployment as all the necessary components (compression and scalable secure aggregation) are ready. On the other hand, discrete Gaussian -- despite being well-explored in the lattice-crypto community over the last decade (e.g. [a, b, c]) -- is still not widely available in numerical analysis packages such as NumPy, TensorFlow, SciPy that ML / DP practitioners would use for development. In the differential privacy context, discrete Gaussian has only recently been introduced (NeurIPS 2020, [12]). In this case, one could either depend on discrete Gaussian and initiate similar development efforts, or simply use Skellam which allows the distributed DP setting for federated learning to be readily deployed as enabled by our work. Please also see our response Reviewer P222 for more details.
>
> [a] Roy, S.S., Vercauteren, F. and Verbauwhede, I., 2013, August. High precision discrete Gaussian sampling on FPGAs. In International Conference on Selected Areas in Cryptography (pp. 383-401). Springer, Berlin, Heidelberg.
>
> [b] Dwarakanath, N.C. and Galbraith, S.D., 2014. Sampling from discrete Gaussians for lattice-based cryptography on a constrained device. Applicable Algebra in Engineering, Communication and Computing, 25(3), pp.159-180.
>
> [c] “Simple, fast and constant-time Gaussian sampling over the integers for Falcon”, 2019. https://csrc.nist.gov/CSRC/media/Events/Second-PQC-Standardization-Conference/documents/accepted-papers/rossi-simple-fast-constant.pdf
>
>
> > “The main strength of this mechanism is that Skellam is closed under summation but DGaussian is not. It is unclear to me that this advantage is enough for publication of this paper.”
>
> From a theoretical standpoint, being closed under summation removes the need for accounting for the divergence errors, which, as measured in DDGauss [19] as an infinity divergence, grow with the number of clients $n$ (as opposed to no such dependence for Skellam), which can be problematic with settings of massively distributed client base (e.g. federated analytics).
>
> Concretely, consider the simple quantile estimation problem [a] where we have a large number of clients ($n \ge 1000$) and sensitivity-1 (binary) queries. With a large central noise stddev $\sigma_c \ge 5$, we can achieve a strong privacy ($\varepsilon \approx \Delta / \sigma_c$), but if this noise is to be added locally (via the distributed DP with SecAgg model), the $\tau$ term of DDGauss (Thm 1 of [19]) would significantly degrade the privacy guarantees because large $n$ means more additive divergence terms and ($\sqrt{n}$) smaller local noise stddev which (exponentially) widens the divergence (the effect of which can be deduced from the left of Figure 2 where DDGauss underperforms at small noise levels; in comparison, Skellam degrades more gracefully with smaller noise stddevs). An specific example in this case is $n=10000$, $\sigma_c = 50$, then local stddev $\sigma_l = 0.5$, and at $\alpha=2$ the RDP of Gaussian is 4e-4 and Skellam is 4.0036e-4, while RDP of DDGauss is > 723 (factor of $>10^6$) because of the sum divergence $\tau$. While scaling can help alleviate this issue, it also introduces additional communication costs. We will provide more discussions in the updated version.
>
> From an engineering perspective, while the divergence term for discrete Gaussian is usually small enough, we still need to keep track of it and its dependent parameters (number of clients, client dimensions, variance) for accurate privacy accounting. Skellam on the other hand only requires us to track the variance and thereby allows easy switching between central DP (noise on the server) vs distributed DP (noise on the clients). We will add this discussion in the revised version of the paper.
>
> [a] “Differentially Private Learning with Adaptive Clipping”, 2019.
>
>
> > “why does PLD section exist if you have at your disposal Theorem 3.5?”
>
> PLD typically provides tighter privacy accounting at the expense of tracking the entire distribution of the privacy loss (as opposed to only its moments as done by Renyi DP). That said, we show that while PLD can be analyzed for $d = 1$ (and it leads to the tightest composition analyses for Skellam), computing it for $d > 1$ is very challenging. This motivates the need for a Renyi DP analysis, which turns out to be very competitive in practice. The motivation to include PLD (even just for $d = 1$) is to initiate the study and understand the best possible bounds.

---

> > ### Comment · Reviewer_p2AR · 2021-08-26
> > **Updating Review**
> >
> > Thank you for replying to all the reviewers questions. I still think efficient implementations of DGaussian will come in the near future but I accept that the paper solves a problem that exists right now so I am raising my score.
> > Adding the discussion on being closed under summation will improve the paper.

---

### Official Review · Reviewer_P222 · 2021-07-16

**Rating:** 7
**Confidence:** 4

**Summary:**

This paper proposed a new discrete differential privacy mechanism based on the difference between two independent poisson random variables, which may be a new attractive alternative of discrete Gaussian mechanism. The authors also carefully proves a sharp privacy guarantee via privacy loss distribution, and renyi divergences. Extensive experiments results on centralized, federated learning, demonstrate that the effectiveness of proposed methods.

**Ethics Review Area:**

["I don’t know"]

**Limitations And Societal Impact:**

Yes.

**Main Review:**

The task of achieving differential privacy on finite machines is important, since continuous mechanism of differential privacy may lead to privacy errors when implemented in floating point.


Strengths:
The paper is highly technical and in particular the properties are rather challenging to prove. All the technical parts seem solid, though I have not fully checked the appendices.

The paper is quite well-written, and I can follow it easily.  I also like that it presents a number of comparisons for the application of their proposed mechanism, including centralized and distributed privacy applications.

Weakness:
It’s better to have a comparison of noise sampling time with discrete Gaussian mechanism to further show the advantages

**Time Spent Reviewing:**

2 hours

---

> ### Author Response · Authors · 2021-08-10
> **Author response to initial review**
>
> We thank the reviewer for the thoughtful comments.
>
> > “It’s better to have a comparison of noise sampling time with discrete Gaussian mechanism to further show the advantages”
>
> We’d like to point out that discrete Gaussian sampling itself has been well explored in the lattice cryptography community (e.g., [a, b, c]), and despite not being experts in lattice cryptography, we expect that the efficient sampling routines (often implemented in C/C++, e.g. [d]) would perform as good as optimized Poisson samplers if compared under a similar setting. However, such efficient discrete Gaussian samplers are not readily available for ML and DP development. We believe that the advantage of Skellam in this regard lies in the wide availability of highly optimized and properly tested Poisson samplers in numerical analysis software packages (e.g. NumPy, SciPy). While it is perceivable that discrete Gaussian samplers will become widely available for ML and DP practitioners in the future, Skellam noise provides a practical advantage as differentially private FL systems with secure aggregation are near deployment today. Further, Skellam is closed under summation, so it leads to less book keeping for computing the overall privacy budget compared to the distributed discrete Gaussian (where one needs to carefully track the divergence between sums of discrete Gaussians and a discrete Gaussian).
>
> Upon the reviewer’s suggestion, we did a preliminary comparison of the sampling time against two existing impls of discrete Gaussian sampler [e, f] available to DP practitioners; [e] is not vectorized and thus 1000x slower than Skellam sampling, and [f] is still up to 40% slower in TF eager execution. We will add more details in the final version.
>
> [a] Roy, S.S., Vercauteren, F. and Verbauwhede, I., 2013, August. High precision discrete Gaussian sampling on FPGAs. In International Conference on Selected Areas in Cryptography (pp. 383-401). Springer, Berlin, Heidelberg.
>
> [b] Dwarakanath, N.C. and Galbraith, S.D., 2014. Sampling from discrete Gaussians for lattice-based cryptography on a constrained device. Applicable Algebra in Engineering, Communication and Computing, 25(3), pp.159-180.
>
> [c] “Simple, fast and constant-time Gaussian sampling over the integers for Falcon”, 2019. https://csrc.nist.gov/CSRC/media/Events/Second-PQC-Standardization-Conference/documents/accepted-papers/rossi-simple-fast-constant.pdf
>
> [d] https://github.com/malb/dgs
>
> [e] https://github.com/IBM/discrete-gaussian-differential-privacy
>
> [f] https://github.com/google-research/federated/blob/master/distributed_dp/discrete_gaussian_utils.py

---

> > ### Comment · Reviewer_P222 · 2021-08-30
> > **Updating Review**
> >
> > Thank you for replying to my questions and other reviewers' questions. I also appreciate that the authors have performed a comparison of the sampling time, which shows the advantages of proposed method. The reason I ask this question is that when I perform the discrete gaussian mechanism in a distributed setting,  the algorithm running is quite slow, especially when the vector dimension is extremely high.
> >
> > Thanks again for the responses, and would like to see more detailed comparisons in the final version.

---

### Official Review · Reviewer_WiXo · 2021-07-16

**Rating:** 6
**Confidence:** 3

**Summary:**

This paper introduced and analyzed the multi-dimensional Skellam mechanism which is a discrete differential privacy mechanism based on the difference of two independent Poisson random variables to the federated learning setting. This paper also studies the experiment performance of such mechanism compared with existing continuous gaussian mechanism.

Contributions are, 1) analyzed the mechanism differentially private behavior in multi-dimensional federated learning setting. 2) experimented its performance and analyzed why this mechanism is suitable for real federated learning practice.


**Limitations And Societal Impact:**

For the potential improvement, In the situation where n and d is large, seems like more bit width is required for the mechanism to reach comparable result with respect to continuous gaussian mechanism. More bit width means heavier communication overheads, so, from the perspective of experimenting, more elaborated discussion about this kind of trade-off is better to be given.

Moreover, the main concern is as the author mentioned, the Skellam mechanism has been studied previously and the main contribution of this paper is providing a tighter analysis based on the Renyi divergence. This is the main reason why I tend to reject the paper.

**Main Review:**

The skellam mechanism is not new, the mechanism exists previously. Considering that implementing DP in real life setting in federated learning requires significantly addressing the communication efficiency, sampling implementation and DP implementation in finite computer, this paper gives a good example.

Quality:
Proving the DP behavior of such mechanism does not contribute much theoretically. However, the experiment is comprehensive and the results are meaningful, the mechanism also bypasses the sampling difficulties of previous mechanisms. The implementation code or the engineering work is solid.

Clarity:
The content arrangement is clear and well organized.

Significance:
The mechanism proposed by this paper bypass the sampling difficulty in previous mechanisms in real-life setting thus making it practically valuable. Experiment result is meaningful, the mechanism shows comparable performance in real federated learning tasks with continuous gaussian mechanism.
-----------------------------------

After reading the authors rebuttal I raise my score. I think the authors provide a good feedback to solve my concerns.

**Time Spent Reviewing:**

8h

---

> ### Author Response · Authors · 2021-08-10
> **Author response to initial review**
>
> Thank you for your comments.
>
> > “Proving the DP behavior of such a mechanism does not contribute much theoretically.”
>
> We respectfully disagree and ask the reviewer to reconsider the proof. Indeed, getting *some* DP guarantee for the mechanism is not difficult, but getting a *tight* analysis which has the same bound (up to lower order additive terms) as the continuous Gaussian RDP is a strong theoretical contribution in our opinion.
>
> To highlight this we begin with an easy sub-optimal analysis -- if one uses the standard well-known first-order approximations for Bessel functions that are commonly used (e.g. the one used in the paper [36] (Valovich and Alda)), it is possible to see that this leads to an additive factor of $\frac{\ell_1\text{-sensitivity}}{\mu}$, as opposed to $\frac{\ell_1\text{-sensitivity}}{\mu^2}$ which we show. This difference is massive especially in high-dimensional settings where $\ell_1$-sensitivity scales by a $\sqrt{d}$ factor and experiments show that in fact the additive term will wash out everything else.
>
> To get the $\frac{\ell_1\text{-sensitivity}}{\mu^2}$ additive term (which is truly negligible), we needed to do a careful second-order analysis stemming from the much tighter bound on the ratio of Bessel functions [32] (Ruiz-Antolín and Segura) stated as Lemma 3.8 in the paper. Beyond inheriting the tight bound, the analysis needs to be carefully second-order at every step as loose approximations lead back to $\frac{\ell_1\text{-sensitivity}}{\mu}$. To achieve this, the analysis splits into multiple cases all provided in the appendix. Due to space limitations, we could not present these nuances of the proof in the main paper. We will clarify all these issues in a revised version.
>
> Further, we believe that our second-order analysis of the product of ratios of Bessel functions is a fundamental one and might be of independent interest outside the scope of the Skellam mechanism.
>
> > “For the potential improvement, in the situation where n and d is large, seems like more bit width is required for the mechanism to reach comparable result with respect to continuous gaussian mechanism.”
>
> We thank the reviewer for bringing this up, and we’ll add more discussion in the updated version. We’d like to emphasize that the bit-width is *per-coordinate*, and for each of the coordinates, the communication cost under the distributed Skellam mechanism depends *logarithmically* on $n$ and $d$ (this dependence is the same as prior works, such as [2, 19 (Theorem 2)]). The cost would thus still be small (compared to the 32-64 bit requirements for proper implementation of continuous Gaussian) even when both $n$ and $d$ are large.
> This logarithmic relationship can be observed from Figure 4 in our paper: with $b=10$ bits per-coordinate, the MSE of Skellam can match the continuous Gaussian; with 10x more clients ($n$ from 100 to 1000), we only needed 4 extra bits to make Skellam match the continuous central Gaussian again. The same observation can be made in the 3rd plot of Figure 4 where $n$ and $d$ are increased simultaneously to 10000 and 2000, where $b=18$ bits suffice. To put these numbers into context, Google’s production next-word prediction models for mobile keyboards [a, b] use $n \le 500$ clients per round, and their production DP language model [c] uses $n = 20000$.
>
> [a] “Federated learning for mobile keyboard prediction”. arXiv:1811.03604.
>
> [b] “Federated Learning for Emoji Prediction in a Mobile Keyboard”. arXiv:1906.04329.
>
> [c] “Training Production Language Models without Memorizing User Data”. arXiv:2009.10031.
>
> > “Moreover, the main concern is as the author mentioned, the Skellam mechanism has been studied previously and the main contribution of this paper is providing a tighter analysis based on the Renyi divergence. This is the main reason why I tend to reject the paper.”
>
> Firstly, we would like to emphasize that, in DP, we believe that a tight analysis of a mechanism is as important as the introduction of the mechanism itself. This is a particular feature of privacy preserving systems as the performance strongly depends on the best bounds (even up to constants or lower-order terms) that we can prove for it. In this regard, while the Skellam mechanism was introduced before as mentioned in the paper, its analysis was quite loose and not usable in practice. We highlight why below.
>
> The existing analysis gives very poor privacy guarantees (even for $d = 1$) under composition compared to the analyses we provide. Please see Figure 1 (the green line is what one gets under the existing analysis where the blue dashed line is what one gets under our analysis) – the gap is huge and it renders the existing analysis unusable in learning settings (where a model is trained over thousands of rounds). After just $T=160$ rounds, the existing direct analysis [36] has almost a 20x larger epsilon ($\varepsilon \approx 133$) than the privacy we get under RDP ($\varepsilon \approx 6.9$), and FL training typically takes thousands of rounds.
> Secondly, the analysis only exists for $d = 1$. For larger $d$ if one performs an extension of the analysis, it will lead to the $\frac{\ell_1\text{-sensitivity}}{\mu}$ additional term as described earlier in addition to worse constants and further losses due to composition. As discussed earlier, addressing this issue is non-trivial.

---

### Official Review · Reviewer_SqyM · 2021-07-17

**Rating:** 6
**Confidence:** 4

**Summary:**

This paper studies a new to DP noise addition mechanism, based on the Skellem distribution.  The benefits of the Skellem distribution include closure under addition, a property lost on discrete Gaussians, and its discrete nature which makes it able to be exactly sampled from with finite machines, unlike continuous Gaussian noise.  Furthermore, this work bounds the increase in the scale of the noise compared with Gaussian noise for the same level of privacy.  Lastly, they provide experimental results of their approach.


**Ethical Concerns:**

This paper deals with rigorous notions of privacy.

**Main Review:**

I like how this paper makes a strong argument for using the Skellem Mechanism rather than the discrete Gaussian distribution.  The federated learning setting seems especially problematic for the discrete Gaussian, as one would like to analyze the utility of several terms added, which is no longer discrete Gaussian.

However, the Skellem Mechanism is not new, as the paper points out in citing [36], but only for the scalar case. Section 3.2 goes into detail for how this paper’s analysis is tighter.

My main criticism of this work also extends to the work on the discrete Gaussian.  The paper from Ilya Mironov gives the typical sampling routine of the Laplace distribution and shows how it can have catastrophic privacy loss, much more than intended.  This work and the work on the discrete Gaussian do not show such an attack on continuous Gaussian noise.  These papers would be much stronger if they showed a particular sampling routine had similar catastrophic privacy loss as the Laplace sampling routine.  Hence, I wish there would be more motivation in this paper for why continuous Gaussian noise should not be used rather than assuming some future attack.

Minor Comments:
- I believe the analysis for the PLD composition requires that the mechanisms are not chosen adaptively.  This applies for the settings you consider, since each coordinate has independent noise, but this limitation should still be explicitly stated.
- “regarding regarding” in line 178.


###########
Update: I have read the author responses and the other reviewers comments.  The application of discrete noise to FL makes sense as it reduces communication cost, so the use case is crucial to demonstrate the need for such types of mechanism like Discrete Gaussian and Skellam.  Thanks for making this clearer, so I will increase my score.  However, there are still issues with presentation, like making it clear the improvement over the existing privacy guarantee for the d = 1 case as was stated in the response to reviewer WiXo.  However, I still do not see the real need for Skellam over Discrete Gaussian.  The 2 stated reasons are that Skellam is closed under addition and there are existing packages for Skellam.  The response to reviewer p2AR states that discrete Gaussian "is still not widely available in numerical analysis packages such as NumPy, TensorFlow, SciPy that ML / DP practitioners would use for development" which is a pretty simple fix.

**Time Spent Reviewing:**

2-3 hours

---

> ### Author Response · Authors · 2021-08-10
> **Author response to initial review**
>
> Thank you for your comments.
>
> > “This work and the work on the discrete Gaussian do not show such an attack on continuous Gaussian noise.”
>
> While dealing with continuous noise distributions is something we’d like to avoid because finite computers cannot exactly represent them -- and therefore floating point implementations often leave a surface for privacy attacks -- we would like to stress that this is not the primary motivation for using discrete distributions in our work. Our target application is the distributed (federated) setting, and here, a discrete distribution is needed for two reasons.
>
> 1. We focus on a federated learning setting with secure aggregation (a cryptographic protocol that computes integer modular sums securely), and thus we cannot use continuous noise distributions with cryptographic protocols and have to consider integer distributions.
>
> 2. Communication is a major bottleneck in federated learning and therefore we’d like to have mechanisms that are private at a communication cost (much) less than the typical 32/64-bit floating point arithmetic (e.g. 16 bits as we show in our work). Our work presents a mechanism that guarantees DP even at low bit widths (perhaps at the cost of lower accuracy). In comparison, the DP guarantees may not hold under the continuous Gaussian if the output is quantized to less than 32 bits.
>
> > “the Skellam Mechanism is not new, as the paper points out in citing [36], but only for the scalar case. Section 3.2 goes into detail for how this paper’s analysis is tighter.”
>
> We would like to emphasize that the existing analysis not only doesn’t extend to the d-dimensional case (see also our response to Reviewer WiXo for details), but also gives very poor privacy guarantees (even for $d = 1$) under composition compared to the analyses we provide.
>
> If we look at Figure 1 (the green line is what one would get under the existing analysis [36] with advanced composition (AC) [17, 22] while the blue line uses our RDP analysis), we can see a huge privacy gap:  after just $T=160$ rounds, the existing direct analysis [36] has almost a 20x larger epsilon ($\varepsilon \approx 133$) than the privacy we get under RDP ($\varepsilon \approx 6.9$). Even when we ignore d-dimensional applications, such gaps render the existing analysis unusable in learning settings where a model is trained over thousands of steps.
>
> More generally, we believe that a tight analysis of a mechanism is as important as the introduction of the mechanism itself because the performance of a privacy preserving system strongly depends on the best bounds that we can prove for it, even up to constants or lower-order terms (see also Figure 12 in the Appendix and our response to Reviewer WiXo).

---

> > ### Comment · Reviewer_SqyM · 2021-08-25
> > **Updating review**
> >
> > I have read the author responses and the other reviewers comments.  The application of discrete noise to FL makes sense as it reduces communication cost, so the use case is crucial to demonstrate the need for such types of mechanism like Discrete Gaussian and Skellam.  Thanks for making this clearer, so I will increase my score.  However, there are still issues with presentation, like making it clear the improvement over the existing privacy guarantee for the d = 1 case as was stated in the response to reviewer WiXo.  However, I still do not see the real need for Skellam over Discrete Gaussian.  The 2 stated reasons are that Skellam is closed under addition and there are existing packages for Skellam.  The response to reviewer p2AR states that discrete Gaussian "is still not widely available in numerical analysis packages such as NumPy, TensorFlow, SciPy that ML / DP practitioners would use for development" which is a pretty simple fix.

---

### Decision · Program_Chairs · 2021-09-27

**Decision:**

Accept (Poster)

**Comment:**

The reviewers were convinced by the technical content of the paper, but were somewhat unconvinced of the value over the distributed discrete Gaussian. The main claimed advantages seemed to be that the primitives were readily available in more libraries (which reviewers did not think was a strong argument), and that the discrete Gaussian is not closed under summation. There is some discussion of this comparison at the end of Section 4. It is requested that the authors place this comparison front-and-center, and expand more upon the points, in order to make the comparison crystal clear. In particular, the authors must be more detailed with their critiques of the discrete Gaussian. What is stopping the discrete Gaussian from being implemented in libraries besides initiative from some interested researcher? In terms of how the discrete Gaussian does not compose: how do you square this with the following sentence from [19]? "The bound of the theorem is surprisingly strong; if σ^2 = τ^2 = 3, then the bound is ≤ 10^{-12}". The plots (Figure 2 and 3) seem to indicate larger differences than what I would have expected from this statement, even in the parameter regime corresponding to this statement. The authors must precisely, thoroughly, and honestly detail why this method is favorable to the discrete Gaussian in the final version. Despite some cynicism about the motivation, this paper had a very thorough analysis of a natural mechanism, and thus should be accepted.